# RTN4IP1 is required for the final stages of mitochondrial complex I assembly and CoQ biosynthesis

Monika Oláhová [ID][1,2,14 ✉], Rachel M Guerra[3,14], Jack J Collier[1,4], Juliana Heidler [ID][5,6,7], Kyle Thompson[1], Chelsea R White[3], Paulina Castañeda-Tamez[5,6], Alfredo Cabrera-Orefice[5,6], Robert N Lightowlers [ID][8], Zofia M A Chrzanowska-Lightowlers [ID][8], Alexander Galkin[9], Ilka Wittig [ID][5,6,10], David J Pagliarini [ID][3,11,12] & Robert W Taylor [ID][1,13 ✉]

## Abstract

**A biochemical deficiency of mitochondrial complex I (CI) underlies approximately 30% of cases of primary mitochondrial disease, yet the inventory of molecular machinery required for CI assembly remains incomplete. We previously characterised patients with isolated CI deficiency caused by segregating variants in *RTN4IP1*, a gene that encodes a mitochondrial NAD(P)H oxidoreductase. Here, we demonstrate that RTN4IP1 deficiency causes a CI assembly defect in both patient fibroblasts and knockout cells, and report that RTN4IP1 is a bona fide CI assembly factor. Complexome profiling revealed accumulation of unincorporated ND5-module and impaired N-module production. *RTN4IP1* patient fibroblasts also exhibited defective coenzyme Q biosynthesis, substantiating a second function of RTN4IP1. Thus, our data reveal RTN4IP1 plays necessary and independent roles in both the terminal stages of CI assembly and in coenzyme Q metabolism, and that pathogenic *RTN4IP1* variants impair both functions in patients with mitochondrial disease.**

**Keywords** Complex I Assembly; Coenzyme Q; Complexome Profiling; Mitochondria; RTN4IP1
**Subject Categories** Metabolism; Molecular Biology of Disease; Organelles

## Introduction

Mitochondria are functionally diverse organelles whose importance in humans is most strongly evidenced by their dysfunction in patients with primary mitochondrial disease (Collier et al, 2023; McBride et al, 2006; Suomalainen and Nunnari, 2024). This group of disorders demonstrates remarkable clinical, genetic, and functional heterogeneity, and is caused by pathogenic variants in genes encoding mitochondrial components (Thompson et al, 2020). The largest group of mitochondrial diseases are caused by variants affecting oxidative phosphorylation (OxPhos) (Fernandez-Vizarra and Zeviani, 2021), the process by which mitochondria synthesise ATP. OxPhos is driven by five complexes (CI-V) embedded within the inner mitochondrial membrane. CI-IV comprise the mitochondrial respiratory chain, which generates the proton motive force coupled to the electron transfer through a series of redox centres, terminating in oxygen. CV harnesses this electrochemical proton gradient to drive ATP production (Tang et al, 2020; Zhao et al, 2019).

OxPhos complexes are composed of multiple subunits and cofactors. Several mitochondrial assembly factors and membrane insertases ensure their faithful formation (Tang et al, 2020). CI (NADH:ubiquinone oxidoreductase), the largest respiratory chain complex, is a ~1 MDa multimeric assembly of 44 different subunits encoded by genes in either the nuclear or mitochondrial genomes and is the primary entry point for electrons, which are transferred from NADH along a series of seven iron-sulphur (Fe-S) clusters to ubiquinone (CoQ) (Hirst, 2013). Biogenesis of CI is a stepwise process involving modular assembly coordinated by at least 18 known assembly factors (Formosa et al, 2018; Stroud et al, 2016; Sung et al, 2024). In humans, CI assembly requires the formation of distinct intermediate pre-assemblies to form a functional enzyme. The Q- and

[1]Mitochondrial Research Group, Translational and Clinical Research Institute, Faculty of Medical Sciences, Newcastle University, Newcastle upon Tyne, UK. [2]Department of Applied Sciences, Faculty of Health & Life Sciences, Northumbria University, Newcastle upon Tyne, UK. [3]Department of Cell Biology and Physiology, Washington University School of Medicine, St. Louis, MO 63110, USA. [4]Department of Clinical Neurosciences, John Van Geest Centre for Brain Repair, University of Cambridge, Cambridge, UK. [5]Center for Functional Proteomics, Faculty of Medicine, Goethe University, 60590 Frankfurt am Main, Germany. [6]Institute for Cardiovascular Physiology, Faculty of Medicine, Goethe University, 60590 Frankfurt am Main, Germany. [7]University Clinic of Vascular Surgery, Innsbruck Medical University, Anichstr. 35, A-6020 Innsbruck, Austria. [8]Mitochondrial Research Group, Biosciences Institute, Faculty of Medical Sciences, Newcastle University, Newcastle upon Tyne, UK. [9]Feil Family Brain and Mind Research Institute, Weill Cornell Medicine, New York 10065 NY, USA. [10]German Center for Cardiovascular Research (DZHK), Partner site RheinMain, Frankfurt, Germany. [11]Department of Biochemistry and Molecular Biophysics, Washington University School of Medicine, St. Louis, MO 63110, USA. [12]Department of Genetics, Washington University School of Medicine, St. Louis, MO 63110, USA. [13]NHS Highly Specialised Service for Rare Mitochondrial Disorders, Newcastle upon Tyne Hospitals NHS Foundation Trust, Newcastle upon Tyne, UK. [14]These authors contributed equally: Monika Oláhová, Rachel M Guerra. ✉E-mail: monika.winter@northumbria.ac.uk; robert.taylor@ncl.ac.uk

N-modules form the hydrophilic matrix arm of the enzyme whilst ND1-, ND2-, ND4- and ND5-modules produce the hydrophobic membrane arm. The insertion of mitochondrial DNA encoded CI subunits (MT-ND1, MT-ND2, MT-ND3, MT-ND4, MT-ND4L, MT-ND5 and MT-ND6) into the membrane arm modules appears to be regulated by specific co-translational systems, involving the mitochondrial insertase OXA1L and the translational regulator of CI, MITRAC15 (Formosa et al, 2020; Guerrero-Castillo et al, 2017; Poerschke et al, 2024; Thompson et al, 2018; Wang et al, 2020). CI assembly is initiated by the union of the proximal membrane arm sub-assemblies, ND1 and ND2, in the inner mitochondrial membrane, and the joining of the Q-module. The distal membrane arm ND4- and ND5-module subcomplexes join with the proximal membrane arm sub-assemblies before the final stages of CI assembly can take place. Finally, the N-module is incorporated by merging with the Q-module, thus completing the peripheral arm of CI (Formosa et al, 2018; Guerrero-Castillo et al, 2017; Wittig and Malacarne, 2021). Fully assembled CI can form respiratory supercomplexes with complexes III$_2$ and IV; however, the physiological role for these supercomplex assemblies continues to be debated (Brischigliaro et al, 2023; Hirst, 2018; Stuchebrukhov et al, 2020; Vercellino and Sazanov, 2021).

CI deficiencies account for ~30% of all primary mitochondrial disease cases (Fassone and Rahman, 2012; Klopstock et al, 2021; Rodenburg, 2016; Stenton and Prokisch, 2020). Genomic and multiomic technologies continue to help identify causal disease-associated genetic variants and delineate the mechanisms regulating CI biology (Floyd et al, 2016; Guerrero-Castillo et al, 2017; Rensvold et al, 2022; Stroud et al, 2016; Sung et al, 2024). Additional molecular genetic investigations of patients with CI deficiency have validated the role of candidate genes in CI assembly, critically shaping our understanding of this process (Alston et al, 2016; Alston et al, 2020; van der Ven et al, 2023). Despite these synergetic approaches, the complete repertoire of proteins required for CI production remains undetermined. Given the predominance of CI deficiencies in causing mitochondrial disease phenotypes, and the reported roles of CI in cancer (Al Assi et al, 2024; Bezwada et al, 2024; Wheaton et al, 2014) and neurodegeneration including Parkinson's disease (Flones et al, 2024; Gonzalez-Rodriguez et al, 2021; Wheaton et al, 2014), it is essential to further improve the mechanistic resolution of CI assembly pathways and CI biology.

We previously identified a series of patients, including one with isolated CI deficiency, presenting with variable neurological phenotypes ranging from isolated optic atrophy to severe early-onset encephalopathy due to pathogenic, bi-allelic, variants in the mitochondrial Reticulon-4-interacting protein 1 (*RTN4IP1*) gene (Charif et al, 2018). Here, we demonstrate that *RTN4IP1* encodes a late-stage CI assembly factor and show that its roles in both CI assembly and CoQ biosynthesis are linked to disease pathology.

# Results

## *RTN4IP1* patient fibroblasts and *RTN4IP1* knockout cells exhibit mitochondrial CI respiratory deficiency

Following the first reported association of bi-allelic, pathogenic *RTN4IP1* gene variants with either isolated optic atrophy or a multisystem neurological disease presentation with optic nerve involvement (Angebault et al, 2015), (*OMIM: 610502*), targeted and whole exome sequencing approaches have been used to identify further disease-causing variants and establish a genotype-phenotype correlation (Angebault et al, 2015; Aldosary et al, 2022; Charif et al, 2018). As part of the *RTN4IP1*-patient cohort reported by Charif and colleagues (2018), we identified a female child harbouring bi-allelic pathogenic variants in *RTN4IP1* (NM_032730.5); a maternally-inherited c.500 C > T, p.Ser167Phe and a de novo c.806 + 1 G > A variant which impaired splicing of *RTN4IP1* transcripts (Fig. 1A and (Charif et al, 2018)). Also known as Optic Atrophy-10 (OPA10), at the time RTN4IP1 was assigned as a mitochondrial uncharacterised protein (MXP) (Floyd et al, 2016) that was later shown experimentally to be important for normal OxPhos function (Arroyo et al, 2016). We showed that these specific *RTN4IP1* variants led to complete loss of immunodetectable RTN4IP1 protein, resulting in a CI assembly defect in patient skeletal muscle (Family 11 in (Charif et al, 2018)).

To further interrogate the role of RTN4IP1 in mitochondrial metabolism, we characterised the OxPhos function in primary *RTN4IP1*-patient fibroblasts carrying c.500 C > T; c.806 + 1 G > A variants and in a CRISPR-Cas9-generated U2OS RTN4IP1 knock-out cell model. SDS-PAGE and immunoblotting showed a steady-state decrease in the CI subunits NDUFB8, NDUFA13 and NDUFA9 in *RTN4IP1* patient fibroblasts. Subunits from OxPhos CII-CV were unaffected compared to controls (Fig. 1B). Next, we investigated the respiratory capacity of *RTN4IP1*-patient fibroblasts, observing a mild decrease (not significant) in basal respiration in patient-derived cells relative to control (Fig. 1C). Oxygen consumption promoted by proton leakage upon CV inhibition was unaffected in *RTN4IP1* patient compared to control fibroblasts, signifying normal integrity of the inner mitochondrial membrane and coupling of respiration with ATP synthesis (Fig. 1C). The electron transport system (ETS) capacity, representing uncoupled respiration at optimum carbonyl cyanide p-trifluoro methoxyphenylhydrazone (FCCP) concentration for maximum respiratory flux, was slightly decreased in the *RTN4IP1* patient, although this was not significant (Fig. 1C).

To further investigate the molecular functions of RTN4IP1, we generated a tractable model cell system (U2OS) lacking RTN4IP1 to facilitate an extended characterisation of RTN4IP1 function. Using CRISPR-Cas9 genome editing, we created heterozygous RTN4IP1$^{KO+/−}$ and homozygous RTN4IP1$^{KO−/−}$ (hereinafter RTN4IP1$^{KO}$) cell lines, as well as isogenic wild-type (WT) controls. RTN4IP1$^{KO+/−}$ and RTN4IP1$^{KO}$ cells showed decreased and completely abolished RTN4IP1 protein levels, respectively (Fig. 1D). Similar to *RTN4IP1*-patient fibroblasts, immunoblotting revealed a marked decrease in the steady-state levels of assessed CI subunits NDUFA9, NDUFB8 and NDUFS1 in RTN4IP1$^{KO}$ cells compared to WT (Fig. 1D,E). Steady-state protein levels of all other assessed OxPhos components were largely unaffected (Fig. 1E). Respirometry measurements in RTN4IP1$^{KO}$ cells revealed a severe decrease in oxidative capacity (Fig. 1F). While the proton leak-linked respiration rate, resulting from the addition of oligomycin was not significantly different between WT and RTN4IP1$^{KO}$, ETS capacity was significantly diminished in the absence of RTN4IP1 compared to WT control (Fig. 1F). To further analyse the underlying mechanism leading to decreased oxygen consumption in cells lacking RTN4IP1, CI- and CII-linked

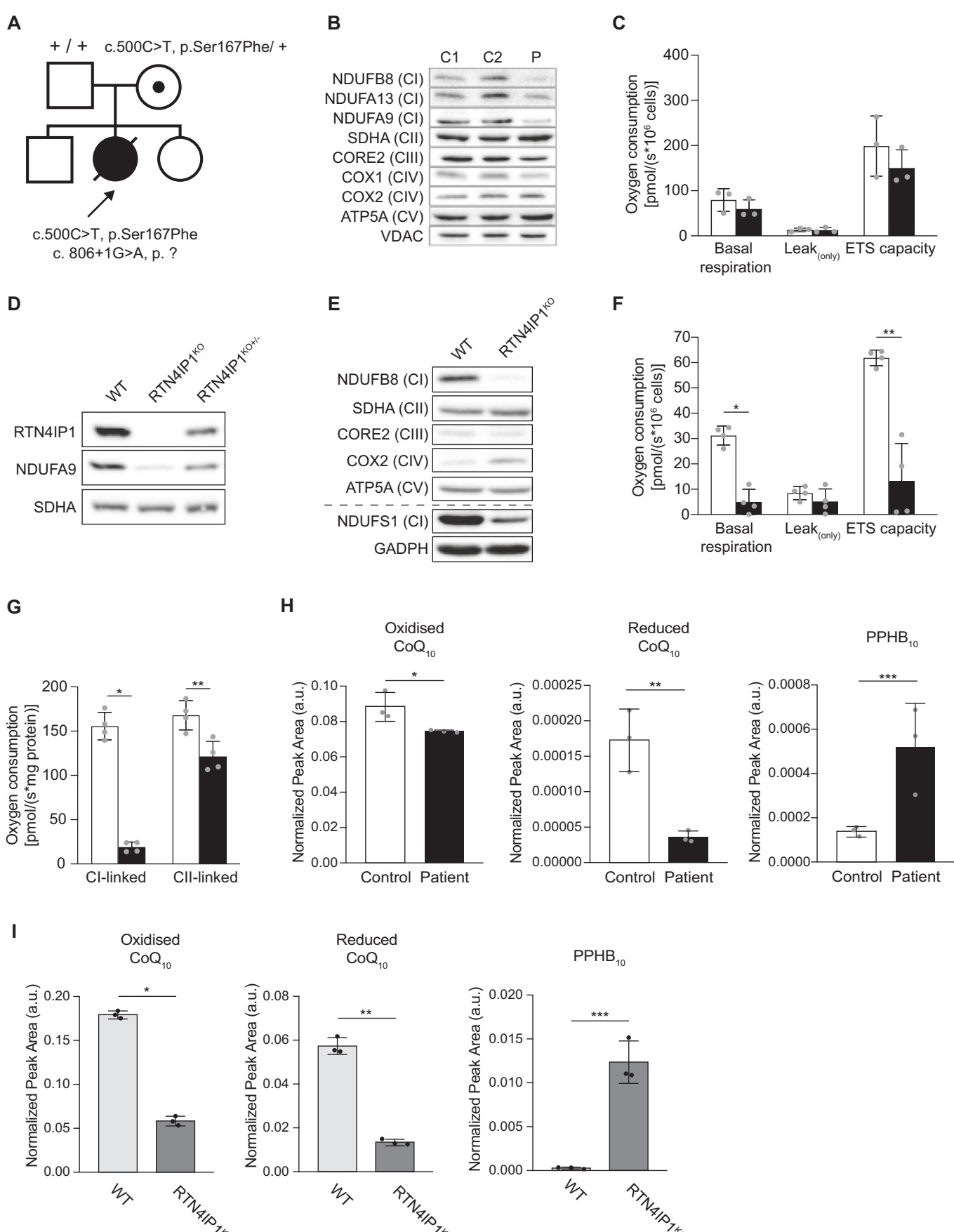

**Figure 1.  RTN4IP1 deficiency impairs CI protein levels, cellular respiration and CoQ biosynthesis.**

(A) Family pedigree showing pathogenic *RTN4IP1* variants. (B) Western blot analysis of protein extracts from control (C1, C2) and *RTN4IP1* patient fibroblasts (P) showing an isolated CI defect. VDAC and SDHA were used as loading controls ($n = 2$). (C) High-resolution respirometry of patient-derived *RTN4IP1* fibroblasts (black bars) compared to control fibroblasts (white bars). ROX-corrected analysis of basal respiration, leak respiration and maximum uncoupled respiration (ETS capacity) are shown. ($n = 3$ mean ± SD). Basal p = 0.352 (ns); Leak $p = 0.883$ (ns) and ETS capacity $p = 0.339$ (ns), two-sided Student's *t*-test. (D) Western blot analysis confirmed the generation of CRISPR-Cas9-edited heterozygous ($+/-$) and homozygous ($-/-$) RTN4IP1$^{KO}$ U2OS lines and isogenic WT control. SDHA was used as a loading control ($n = 2$). (E) Western blot analysis of OxPhos complex subunits, showing an isolated CI defect in RTN4IP1$^{KO}$ compared to WT. GAPDH and SDHA were used as loading controls ($n = 3$). (F, G) Respirometry analysis as described in (C) in U2OS WT (white bars) and RTN4IP1$^{KO}$ (black bars) cells showing a respiration defect caused by a severe reduction of CI-linked respiration. ($n = 4$ mean ± SD). In (F) Basal *$p = 0.000157$, Leak $p = 0.280$ (ns) and ETS capacity **$p = 0.000677$. In (G) CI-linked respiration *$p = 3.23e{-}6$ and CII-linked respiration **$p = 0.00773$, two-sided Student's *t*-test. (H, I) Targeted lipidomic analysis on (H) *RTN4IP1* patient (P) and control (C) fibroblasts and (I) RTN4IP1$^{KO}$ and WT U2OS cells, ($n = 3$ mean ± SD). In (H), *$p = 0.0365$, **$p = 0.00603$, ***$p = 0.0287$; in (I), *$p = 6.68e{-}6$, **$p = 4.81e{-}5$, ***$p = 0.000991$; two-sided Student's *t*-test. Source data are available online for this figure.

respiratory rates were measured, showing significant reductions of >88% and ~28%, respectively. Our collective data point towards CI as the precise site responsible for this observed decrease in respiration (Fig. 1G).

## Pathogenic *RTN4IP1* variants lead to an impairment in coenzyme Q biosynthesis

Previous characterisation of protein-protein interactions by affinity enrichment mass spectrometry (AE-MS) of 50 MXPs (Floyd et al, 2016) captured an interaction between RTN4IP1 and COQ9, an auxiliary lipid-binding protein involved in the CoQ biosynthesis pathway (Guerra and Pagliarini, 2023). Furthermore, recently published work has characterised RTN4IP1 as a mitochondrial matrix oxidoreductase that supports coenzyme Q biosynthesis, specifically through regulation of the O-methyltransferase activity of COQ3 (Park et al, 2024). We therefore performed a targeted lipidomic analysis of *RTN4IP1*-patient fibroblasts, which revealed a mild loss of oxidised coenzyme Q10 (CoQ$_{10}$), and a more striking loss of reduced CoQ$_{10}$ compared to control fibroblasts, which was accompanied by elevated polyprenyl-hydroxybenzoate 10 (PPHB$_{10}$), an early CoQ biosynthesis pathway intermediate that typically accumulates when the CoQ pathway is defective (Fig. 1H). Targeted lipidomics of both WT and RTN4IP1$^{KO}$ cells revealed a significant decrease in both oxidised and reduced CoQ$_{10}$ levels, accompanied by an increase in PPHB$_{10}$ (Fig. 1I). The marked loss of CoQ$_{10}$, together with the concomitant accumulation of an early biosynthetic intermediate, is consistent with impaired biosynthesis, suggesting a role for RTN4IP1 in regulating this pathway.

Analysis of a previously published large-scale multi-omics study (Rensvold et al, 2022) that included the knockout collection of genes encoding for CoQ biosynthetic enzymes and CI structural subunits and assembly factors, allowed us to interrogate any reciprocal relationship between CoQ and CI deficiencies. While the COQ$^{KO}$ lines exhibited an expected significant loss of total CoQ$_{10}$ levels, no perturbation of CoQ$_{10}$ levels was observed in the CI subunit and assembly factor KOs (Fig. EV1A). Additionally, analysis of the proteomics data for the COQ$^{KO}$ cell lines (COQ2, COQ7 and COQ8A/B) did not cause a global decrease in CI subunit abundance (Fig. EV1B). These data indicate that while CoQ biosynthesis and CI assembly are intricately linked processes, disruption of one does not automatically lead to the impairment of the other, supporting a possible dual role for RTN4IP1 in these pathways.

## RTN4IP1 deletion selectively impairs CI biogenesis

Given the observed decrease in CI subunits in RTN4IP1-deficient cells, we next investigated how loss of RTN4IP1 affected the assembly of OxPhos complexes by analysing *N*-dodecyl β-D-maltoside (DDM)-solubilised mitochondrial membrane extracts from WT and RTN4IP1$^{KO}$ using blue-native electrophoresis (BNE) and immunoblotting analysis. Applying antibodies against the N-module subunit of CI, NDUFV1, we were unable to detect fully assembled CI in RTN4IP1$^{KO}$ compared to WT, indicating a key role for RTN4IP1 in CI biogenesis (Fig. 2A), which is in line with the decrease in steady-state CI subunits in the RTN4IP1$^{KO}$ following immunoblotting analysis (Fig. 1D,E).

Next, we solubilised mitochondrial membranes from WT and RTN4IP1$^{KO}$ cells with digitonin. BNE analysis of Coomassie-stained native gels showed a clear absence of supercomplex S$_1$ (I-III$_2$-IV) in RTN4IP1$^{KO}$ compared to WT (Fig. 2B, left) and undetectable S$_1$ when stained for NADH dehydrogenase activity (Fig. 2B, middle). Moreover, haem staining showed that the pattern of CIV was not affected by loss of RTN4IP1 (Fig. 2B, right). As expected, digitonin-solubilised mitochondrial membranes from *RTN4IP1*-derived patient fibroblasts also showed a clear reduction of the CI-containing supercomplex S$_1$ compared to control following BNE (Fig. 2C).

To define the RTN4IP1-associated OxPhos defect with higher resolution, we performed complexome profiling on the patient-derived cells and RTN4IP1$^{KO}$ cells (Heide et al, 2012). Mitochondria were solubilised with digitonin, followed by protein separation by BNE, fractionation and analyses by quantitative MS as previously reported (Fuhrmann et al, 2018). Complexome profiling data were analysed to correlate differences in the abundance and arrangement of multiprotein complexes—from high (left) to low (right) molecular mass—in the form of a "heatmap", focussing specifically on the inventory of subunits and assembly factors of mitochondrial OxPhos components in *RTN4IP1* patient fibroblasts and RTN4IP1$^{KO}$ cells and corresponding controls (Figs. 2D, 3, 4 and EV2–EV4). Consistent with the BNE gel data, supercomplex assembly (predominantly S$_1$; I-III$_2$-IV) was attenuated in *RTN4IP1*-patient cells (Figs. 2D and EV2), whilst a complete loss of fully assembled supercomplexes was noted in RTN4IP1$^{KO}$ cells (Fig. 3A,B). Focussing on the RTN4IP1$^{KO}$ cells, which demonstrated a more dramatic loss of CI-containing supercomplexes and CI-subcomplexes compared to *RTN4IP1* patient-derived fibroblasts, we observed diminished amounts of supercomplex S$_1$ and the presence of a large intermediate of CI (L$_{int}$) (Fig. 3B, orange box). This resulted in a concomitant increase of free CIII and CIV

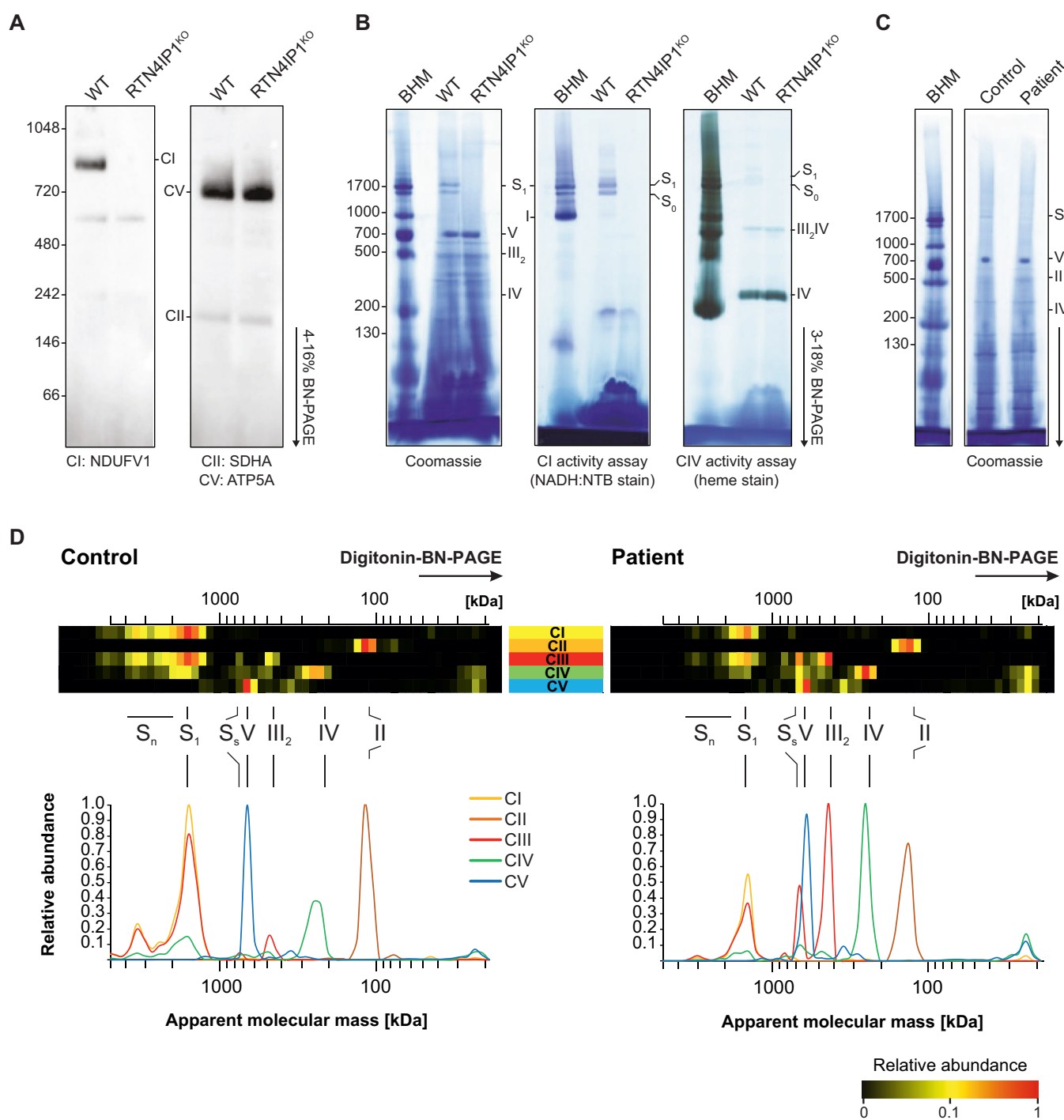

**Figure 2.  RTN4IP1 deficiency selectively impairs CI biogenesis.**

(**A**) BNE and immunoblotting analysis of DDM-solubilised mitochondrial membranes from WT and RTN4IP1[KO] U2OS lines using antibodies against complexes I (NDUFV1), II (SDHA, loading control) and V (ATP5A), (*n* = 3). (**B**, **C**) In (**B**) deletion of RTN4IP1 results in the absence of supercomplex and undetectable NADH:NTB reductase activity. Digitonin solubilisation of mitochondrial membranes from U2OS WT and RTN4IP1[KO] cells separated on native gradient gels were stained with Coomassie (left), NADH:NTB reductase activity stain (middle) and CIV-specific haem stain (right). Representative gels for NADH:NTB and haem stain are shown (*n* = 3). In (**C**) *RTN4IP1* variants cause a reduction of CI-containing supercomplexes. Enriched mitochondrial membranes from control and the *RTN4IP1* patient fibroblasts were solubilised with digitonin and separated on a native gradient gel. Protein complexes were stained with Coomassie (*n* = 3). (**D**) Summary of OxPhos complexes from complexome profiling of control and *RTN4IP1* patient fibroblasts shown as 2D OxPhos complex profiles. Subunits of each individual OxPhos complex were summed up and normalised to the maximum between gel lanes of control and patient samples and illustrated in a heatmap. Assignment of complexes in (**B**, **D**): complex I (I); complex II (II); complex III dimer (III$_2$); complex IV (IV); complex V (V); supercomplex containing CI, III$_2$ and 1 copy of CIV (S$_1$); S$_1$ and extra copies of CIV (S$_{1+n}$); higher order supercomlexes (S$_n$). BHM bovine heart mitochondria solubilized with digitonin served as native mass ladder. Source data are available online for this figure.

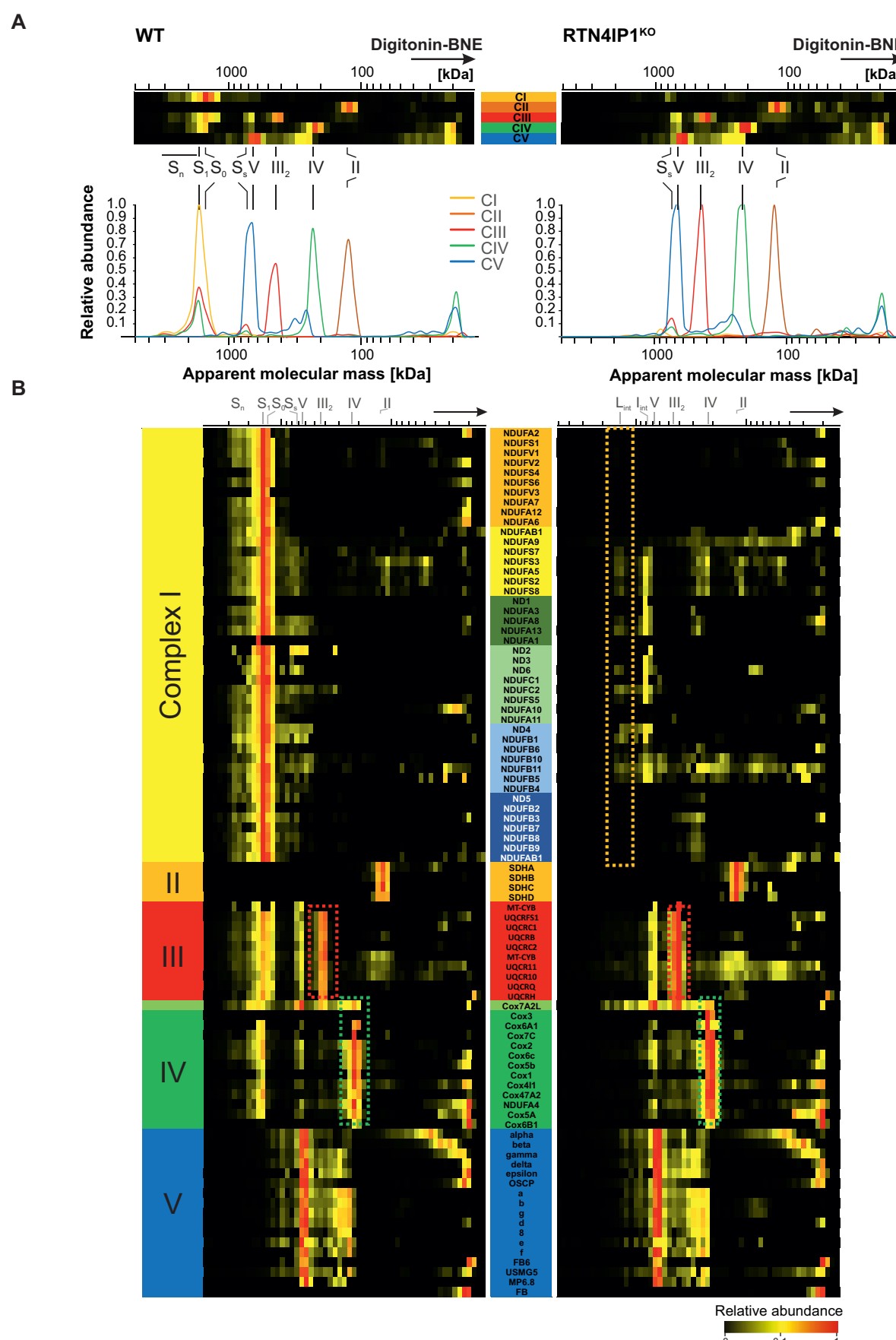

◄

**Figure 3. Complexome profiling analysis of OxPhos complexes from RTN4IP1$^{KO}$ and WT U2OS cells.**

(A) 2D OxPhos complex profiles from complexome data from RTN4IP1$^{KO}$ corresponding to WT cells. Complexome profiling data of OxPhos complexes I-V are presented as 2D plots. (B) OxPhos complexes of RTN4IP1$^{KO}$ and WT cells analysed by complexome profiling. Complexome profiling data were presented as a heatmap, corresponding to subunits of individual OxPhos complexes I-V. In (A, B) Mitochondrial complexes were solubilized with digitonin, separated by BNE, and gel slices were analysed by quantitative MS. Assignment of complexes in (A, B): complex I (I); complex II (II); complex III dimer (III$_2$); complex IV (IV); complex V (V); higher order supercomplexes (S$_n$); supercomplex containing CI, III$_2$ and 1 copy of CIV (S$_1$); supercomplex containing CI and a dimer of CIII (S$_O$); small supercomplex S$_O$ of CIII$_2$ and CIV (S$_s$) and large intermediate CI (L$_{int}$) and intermediate of CI including modules Q, ND1m, ND2m, ND4m and assembly factors (I$_{int}$). The relative abundance of each protein was represented from low to high according to the colour scale illustrated on the bottom right.

(Fig. 3B, red box and green box, respectively). Similar increases in CIII and CIV were also observed in *RTN4IP1*-patient fibroblasts (Fig EV2, red box and green box, respectively), whilst assembly of CII and CV were largely unaffected (Fig. EV2). Collectively, these data support RTN4IP1 having an essential and specific role in CI assembly.

## RTN4IP1 is a late-stage CI assembly factor

To position RTN4IP1 within the stepwise CI biogenesis process, which involves the tightly coordinated assembly of intermediate modules (ND1, ND2, ND4, ND5, Q and N) and cofactors (e.g., FMN, Fe-S clusters), we further interrogated the complexome profile of CI. Analysis of the complexome profile of both RTN4IP1$^{KO}$ cells and *RTN4IP1* patient fibroblasts (Figs. 4 and EV3) demonstrated that while the initial Q-module assembly appeared to be normal, its subsequent sub-assemblies containing membrane subunits clearly accumulated in the absence of RTN4IP1. We also observed the presence of the Q-module assembly factors NDU-FAF3, NDUFAF4 and TIMMDC1, which did not dissociate from the large intermediates (Fig. 4). ND1-module assembly was normal in RTN4IP1$^{KO}$ cells and accumulated together with the Q- and mitochondrial Complex I assembly (MCIA) complex, which is important for the stability of mtDNA-encoded ND2 (Formosa et al, 2020). Similarly, ND2-module assembly appeared normal as the ND2-module was found to comigrate with Q- and ND1-modules and the MCIA complex (NDUFAF1, ACAD9, ECSIT, TMEM126B, TMEM186) (Fig. 4). Although the ND4-module assembled and joined with the Q-, ND1-, ND2-modules and the MCIA complex, it accumulated in sub-assemblies in RTN4IP1-deficient cells, involving at least four ND4-module subunits (NDUFB5, NDUFB6, NDUFB10 and NDUFB11) (Fig. 4, light blue box) and the assembly factors TMEM70 and TMEM126A (Fig. 4, white box). It is noteworthy that WT cells also exhibited the presence of a CI intermediate (I$_{int}$), which encompasses the modules Q/ND1/ND2/ND4 in conjunction with assembly factors. We observed that FOXRED1 and DMAC2, which was recently identified together with other assembly factors (TMEM70, DMAC1, TMEM126A) in ND4 intermediates (D'Angelo et al, 2021; Formosa et al, 2021; Guerrero-Castillo et al, 2017; Stroud et al, 2016), binds to I$_{int}$ in WT, but both appear to be absent in the intermediate I$_{int}$ and L$_{int}$ of RTN4IP1-deficient cells (Fig. 4, white box), suggesting that the absence of RTN4IP1 may disrupt the function of FOXRED1 and DMAC2 in late CI membrane arm assembly. Additional analysis of a DDM-solubilised complexome profile (Fig. EV4) revealed that the most dramatic changes observed in the absence of RTN4IP1 were related to the assembly of ND5-module, which assembled only as a single intermediate (Fig. 4A, dark blue box) or to some extent together with pre-assembled ND4-module (Fig. EV4) together with

the assembly factors TMEM126A, DMAC1 and FOXRED1 but was not able to join the large Q/ND1/ND2-intermediate (Figs. 4, dark blue box and EV4).

Additionally, we were unable to identify the subunits NDUFB4 (ND4-module) and NDUFB2 (ND5-module) (Figs. 4 and EV4). NDUFB4 seems to play a role in supercomplex formation or stability (Parmar et al, 2024). Interestingly, for the ND2 module, signals from NDUFA10 and NDUFA11 were also absent. The latter is an interface subunit to CIII and is likely involved in the stabilisation or assembly of respiratory supercomplexes. We observed an accumulation of N-module subunits (NDUFV1, NDUFV2, NDUFS1 and NDUFA2) and assembly factors (NDU-FAF6, NDUFAF7) at higher molecular masses (60–250 kDa) in the RTN4IP1$^{KO}$ cells and control cells, leading us to hypothesise that the initial formation of the N-module takes place in RTN4IP1$^{KO}$ conditions (Fig. EV4).

## RTN4IP1 deficiency impairs CI assembly in mitochondrial disease

Similar to RTN4IP1$^{KO}$ cells, we found increased Q-module sub-assemblies retaining assembly factors and accumulating in *RTN4IP1*-patient fibroblasts when compared to controls. In addition, a comparable large assembly intermediate I$_{int}$ including Q/ND1/ND2/ND4-modules—just as described in the RTN4IP1$^{KO}$ cells—was present in the *RTN4IP1* patient cells (Fig. EV3). In contrast to RTN4IP1$^{KO}$ cells, we observed that the ND5-module and the final N-module do assemble to complete CI and integrate into supercomplexes. However, we believe this could be the limiting step in this process, as all residual sub-assemblies (I$_{int}$) appear to accumulate with their respective assembly factors (Fig. EV3). Finally, RTN4IP1 was only identified as an individual protein that did not migrate with any of the CI-pre-assemblies when digitonin treated mitochondrial membranes were separated by BNE and analysed by complexome profiling (Figs. 4 and EV3). Collectively, our data converge to show that RTN4IP1 is a CI assembly factor required for the terminal stages of CI assembly.

## RTN4IP1 independently regulates both CI biogenesis and CoQ biosynthesis

Given the severe CI-related defects documented in the RTN4IP1$^{KO}$ line (Fig. 1D,E), in addition to those described in *RTN4IP1*-patient tissues (Fig. 1B) and (Charif et al, 2018), we performed proteomic analyses of WT and RTN4IP1$^{KO}$ cells. To further define the role of RTN4IP1 in CI and CoQ biology, we reintroduced RTN4IP1 into RTN4IP1$^{KO}$ cells and analysed the subsequent phenotypes. As a proposed oxidoreductase, human RTN4IP1 contains a Rossmann fold (GxxGxx(G/A)) that is required for NAD(P)H binding. To

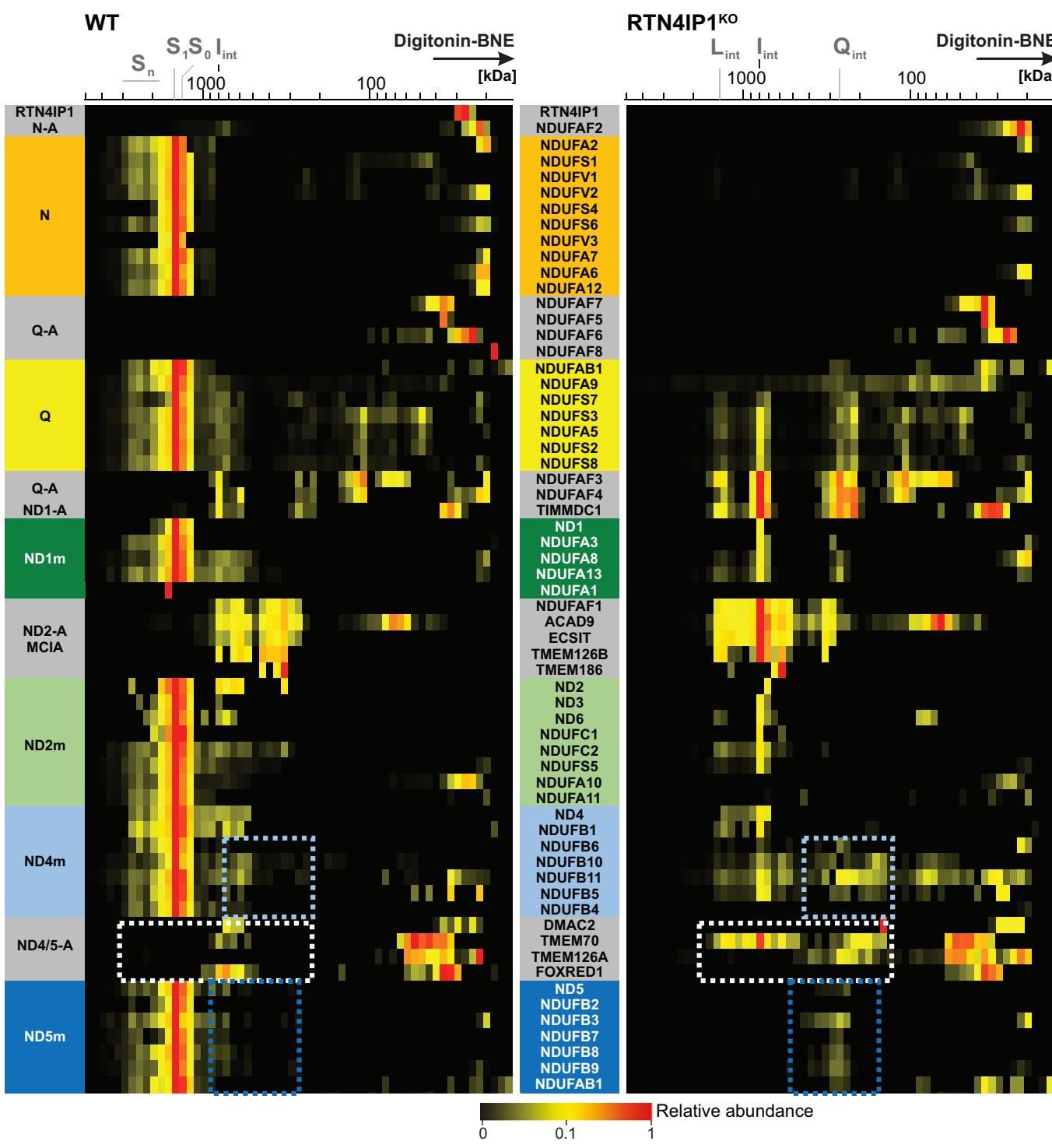

**Figure 4. RTN4IP1 deletion impairs ND5 module attachment and N-module stability in the late stages of CI assembly.**

Complexome profiling data focused on CI and its known assembly factors (marked with *-A) were grouped according to CI assembly modules: N, Q, ND1m, ND2m, ND4m and ND5m. The data were presented as a heatmap. CI-related assemblies are labelled as follows: Sn – higher-order supercomplexes, S1 – supercomplex containing CI, CIII$_2$ and CIV, S0 – supercomplex with CI and dimeric CIII, Lint large CI intermediate including modules Q, ND1m, ND2m, ND4m, and associated assembly factors, categorised by their interaction (N-A, Q-A, ND1-A, ND2-A, ND4/5-A, MCIA). Q$_{int}$ intermediate of the Q-module containing specific Q-associated assembly factors. Protein abundance is colour-coded from low to high as indicated by the scale at the bottom. Highlighted areas: Light blue box: stalled intermediate of the ND4-module, Dark blue box: ND5 module intermediate, White box: ND4/ND5-associated assembly factors.

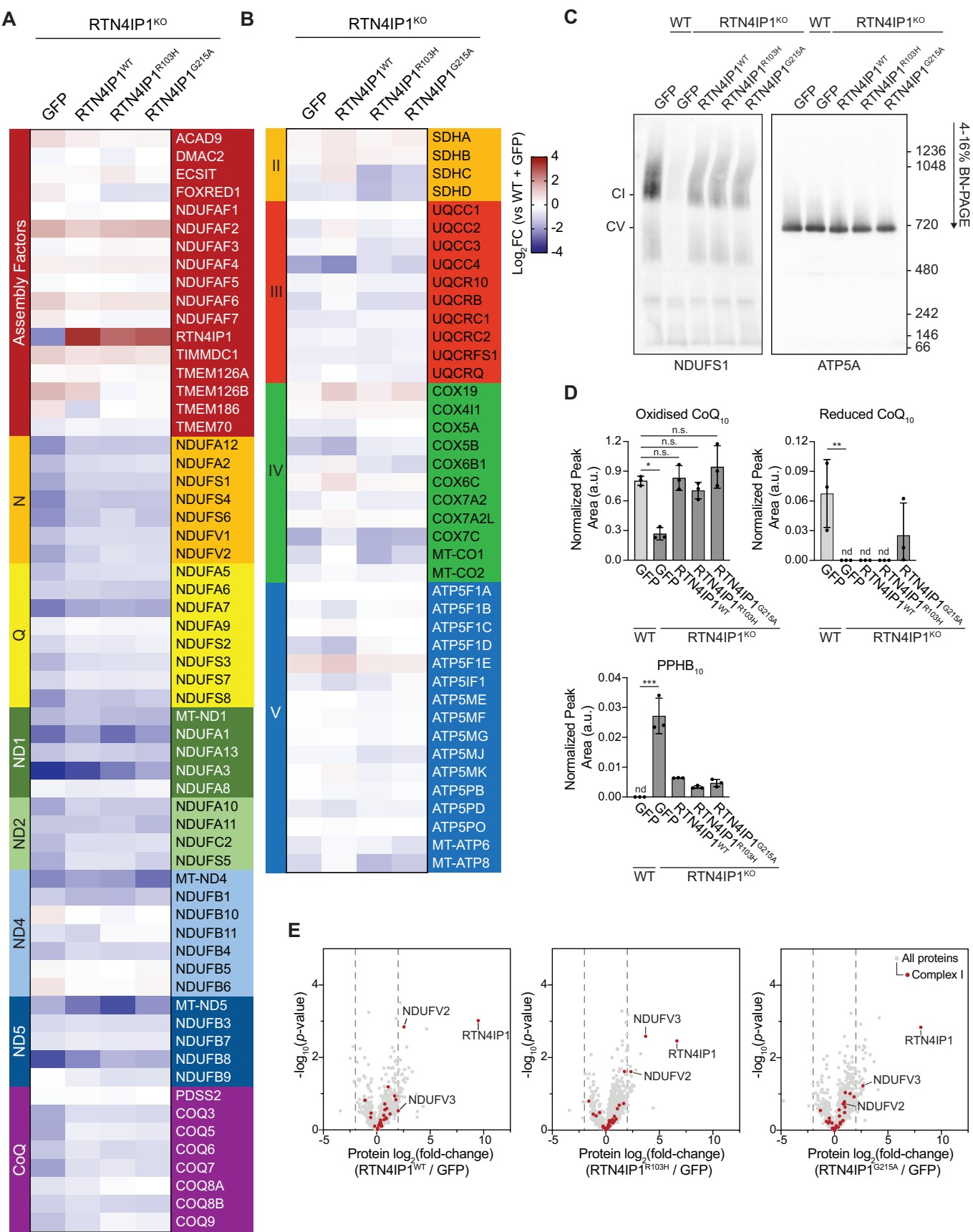

**Figure 5.  Reintroduction of RTN4IP1 restores CoQ production, but only partially rescues CI assembly.**

(A, B) Heatmap depicting the mean $\log_2$(fold change) of CI (A) and OxPhos complex II-V (B) subunits in the RTN4IP1$^{KO}$ cell lines stably expressing the indicated constructs compared to the WT(+GFP) control cell line ($n = 3$). (C) BNE and immunoblotting analysis of DDM-solubilised mitochondrial membranes from WT and RTN4IP1$^{KO}$ U2OS cell lines stably expressing the indicated constructs, using antibodies against CI (NDUFS1) and CV (ATP5A, loading control). (D) Targeted lipidomic analysis on U2OS WT and RTN4IP1$^{KO}$ cells stably expressing the indicated constructs, showing a complete restoration of oxidised CoQ$_{10}$ and PPHB$_{10}$ levels in the RTN4IP1-deficient cells expressing RTN4IP1-FLAG WT, R103H, or G215A. Levels of reduced CoQ$_{10}$ are not rescued with any of the RTN4IP1 constructs. Data shown as mean ± SD ($n = 3$). *$p = 0.000283$, **$p = 0.0271$, ***$p = 0.00139$, n.s. not significant, two-sided Student's *t*-test. (E) Volcano plot showing the mean $\log_2$(fold change) and the $-\log_{10}$ of the two-tailed Student's *t*-test. *P* value (no adjustment for multiple comparisons) of proteins in mitochondria isolated from RTN4IP1$^{KO}$ cells stably expressing the indicated FLAG-tagged RTN4IP1 construct compared to wild-type WT(+GFP) mitochondria: $n = 3$ biologically independent samples per condition. Dashed lines mark $\log_2$(fold change) values of −2 and 2. CI subunits and assembly factors are marked as red dots. All other proteins are marked as grey dots. Source data are available online for this figure.

disrupt this binding site, we generated U2OS RTN4IP1$^{KO}$ cell lines stably expressing two of the catalytically inactive RTN4IP1 mutants (RTN4IP1$^{R103H}$ and RTN4IP1$^{G215A}$) originally identified in the (Park et al, 2024) study (Fig. EV5A). Next, we performed proteomic analysis which confirmed our previous findings (Figs. 1D,E and 3), showing that absence of RTN4IP1 causes a decrease in the abundance of CI subunits (Figs. 5A and EV5B), whilst the levels of CII, CIII, CIV and CV components are largely unaffected (Fig. 5B). Consistent with this, gene ontology overrepresentation analysis of the full proteomic profile of RTN4IP1$^{KO}$ cells also revealed enrichment of factors involved in NADH dehydrogenase complex assembly (GO: 0010257) among the significantly down-regulated proteins (Fig. EV5C). Stable overexpression of WT and mutant RTN4IP1 (RTN4IP1$^{WT}$, RTN4IP1$^{R103H}$ or RTN4IP1$^{G215A}$, respectively) restored the levels of most downregulated CI subunits in the RTN4IP1$^{KO}$ cell line closer to WT levels (Figs. 5A and EV5A,B), confirming that defective RTN4IP1 causes CI deficiency. We next investigated whether reintroduction of either WT or mutant RTN4IP1 (R103H and G215A) can rescue the CI assembly defect by analysing DDM-solubilised mitochondrial membrane extracts by BNE and immunoblotting against the N-module subunit of CI, NDUFS1 (Fig. 5C). All rescue cell lines exhibited a similar pattern, with loss of assembled CI in the RTN4IP1$^{KO}$ cells, and partial rescue with the WT and R103H and G215A mutant RTN4IP1 proteins, suggesting that RTN4IP1's enzymatic activity may not be important for "structural" CI assembly (Fig. 5C).

To further probe the role of RTN4IP1 in CoQ biosynthesis, we performed targeted lipidomic analysis on the cell lines expressing WT and mutant RTN4IP1 proteins. Stable overexpression of RTN4IP1$^{WT}$, RTN4IP1$^{R103H}$ and RTN4IP1$^{G215A}$ respectively, fully rescued the oxidised CoQ$_{10}$ deficiency and accumulation of the biosynthetic intermediate PPHB$_{10}$, however, all failed to rescue the levels of reduced CoQ$_{10}$ (Fig. 5D). Previously reported COQ3 activity assays (Park et al, 2024) with addition of RTN4IP1$^{R103H}$ and RTN4IP1$^{G215A}$, demonstrated that these mutations only marginally impacted the ability of RTN4IP1 to assist in the methylation activity of COQ3, suggesting that RTN4IP1$^{R103H}$ and RTN4IP1$^{G215A}$ mutants still retain some catalytic activity or support the activity of COQ3 in a manner independent of its oxidoreductase activity. We detected most COQ biosynthetic proteins in our proteomics analysis and found that they decreased in abundance in the RTN4IP1$^{KO}$ cells; however, this phenotype was rescued by re-expression of any of the RTN4IP1 forms—RTN4IP1$^{WT}$, RTN4IP1$^{R103H}$ or RTN4IP1$^{G215A}$ (Fig. 5A), further underscoring the physiological role of RTN4IP1 in supporting efficient CoQ$_{10}$

biosynthesis. To gain deeper insights into the molecular role of RTN4IP1 in CI and CoQ biogenesis, we performed affinity enrichment mass spectrometry (AEMS) following disuccinimidyl sulfoxide (DSSO) crosslinking on isolated mitochondria from rescue cell lines expressing either RTN4IP1$^{WT}$, RTN4IP1$^{R103H}$ or RTN4IP1$^{G215A}$, allowing us to probe the interacting partners of RTN4IP1 (Fig. 5E). Previous studies reported an interaction between RTN4IP1 and COQ3, proposing that RTN4IP1 assists COQ3 in the *O*-methylation of its substrate, demethyl-coenzyme Q (DMeQ) (Park et al, 2024). However, we could not detect any interactions between RTN4IP1 and CoQ biosynthetic proteins via AEMS, likely due to technical limitations of the method in capturing weak and/or transient interactions. Instead, the AEMS identified interactions between RTN4IP1 and CI N-module subunits NDUFV2 and NDUFV3 proteins (Fig. 5E). These interactions were identified in cells expressing RTN4IP1$^{WT}$ and RTN4IP1$^{R103H}$ and to a lesser extent in the RTN4IP1$^{G215A}$ expressing cells, particularly with NDUFV2. Together, these data suggest that RTN4IP1 may indirectly stabilise early N-module subcomplexes of the peripheral arm to enable proper maturation of the CI holoenzyme.

In summary, although wild-type and mutant RTN4IP1 (R103H and G215A) fully restore the CoQ biosynthetic defect in the absence of RTN4IP1, they only partially rescue the CI protein abundance and assembly defect (Fig. 5A–D). These results suggest that CI remains functionally impaired and is therefore unable to fully reduce CoQ$_{10}$, which accumulates in its oxidised form. Collectively, these findings indicate that RTN4IP1 is likely playing distinct roles in both CI maturation and CoQ biosynthesis.

## Discussion

Here, we report a critical role for RTN4IP1 in the late-stage assembly of mitochondrial CI and confirm its role in regulating CoQ$_{10}$ biosynthesis in humans. Our interest in RTN4IP1 started with the investigation and subsequent molecular diagnosis of a patient with bi-allelic *RTN4IP1* variants who presented with a severe and isolated CI biochemical deficiency in muscle and fibroblasts (Charif et al, 2018). A closer inspection of the clinical phenotypes associated with primary CoQ$_{10}$ deficiency and patients with *RTN4IP1*-related mitochondrial disease—and other CI-driven pathologies—reveals significant overlap.

Recessively inherited, pathogenic variants in the *RTN4IP1* gene were first identified in several families with isolated optic atrophy (Angebault et al, 2015), leading to the characterisation of larger patient

cohorts with variable neurological phenotypes, thus expanding the clinical phenotype of *RTN4IP1*-related mitochondrial disease (Charif et al, 2018). Optic atrophy and optic neuropathy are commonly associated with other mitochondrial disease pathologies implicating CI deficiency, including Leber hereditary optic neuropathy (Carelli et al, 2023) as well as recessively-inherited variants in structural components of CI such as NDUFS6 (Gangfuss et al, 2024), the CI assembly factor TMEM126A (Formosa et al, 2021) and DNAJC30 which mediates CI repair (Stenton et al, 2021). It is interesting to note that some patients with inherited pathogenic variants in genes implicated in primary $CoQ_{10}$ deficiency manifest with optic atrophy as a feature of complex neurological phenotypes, including mutation of COQ1/PDSS1 (Mollet et al, 2007; Nardecchia et al, 2021), COQ2 (Stallworth et al, 2023) and COQ6 (Justine Perrin et al, 2020). We demonstrated that, in addition to CI deficiency, the CoQ biosynthesis pathway is also impaired in the *RTN4IP1* patient fibroblasts (Fig. 1H). Recent studies, using immortalised C2C12 *RTN4IP1-KO* mouse myoblasts and a muscle-specific *dRTN4IP1-KD* fruit fly model, have shown that $CoQ_2$ supplementation was unable to fully rescue the respiratory defect caused by RTN4IP1 deficiency (Park et al, 2024), suggesting that the function of RTN4IP1 in CI assembly may be distinct from its role in $CoQ_{10}$ biosynthesis, at least in mice and flies.

The biogenesis of CI is a complex and precisely orchestrated process, involving the assembly of pre-produced modules with the aid of specific CI assembly factors. We now identify RTN4IP1 as a novel CI assembly factor that completes the late stages of mitochondrial CI assembly. Tracking the late-acting CI assembly factors of the distal membrane part ND4/ND5-module bound to intermediates (Fig. 4, white box), we found that the CV assigned assembly factor TMEM70 (Carroll et al, 2021; Sanchez-Caballero et al, 2020) and TMEM126A (Formosa et al, 2021) are still involved in late-stage assembly intermediates in the absence of RTN4IP1. However, DMAC2 (Stroud et al, 2016) and FOXRED1 (Formosa et al, 2015) have dissociated, as they are not found in high molecular mass intermediates and were only identified at the low molecular mass. Although the role of FOXRED1 in the late assembly of the membrane portion is established through the study of *FOXRED1*-deficient cells, its specific molecular function as a putative FAD-dependent oxidoreductase is not completely elucidated (Formosa et al, 2015; Hock et al, 2020). Based on our data, we hypothesise that the mitochondrial matrix NAD(P)H oxidoreductase RTN4IP1 has a function at this point of CI assembly, as the termination products are found after the assembly of the ND4-module. Despite an exhaustive examination of the complexome data, deposited in the PRoteomics IDEntifications (PRIDE) Archive database (Perez-Riverol et al, 2022), RTN4IP1 was not identified in the high molecular mass region of the late assembly intermediates. This strongly suggests that RTN4IP1 may bind to the complex in a loose and transient manner, acting as a soluble matrix protein to fulfil its function as a late CI assembly factor. It remains unclear whether RTN4IP1 directly interacts with the ND4- or ND5-modules, or whether it functions as an oxidoreductase with FOXRED1, allowing the joining of ND4- and ND5-modules and signalling the readiness for N-module docking. However, enrichment experiments (Fig. 5E) revealed interaction with NDUFV1 and NDUFV2 and in DDM-complexomes, RTN4IP1 appears in control condition in higher molecular masses that comigrate with preformed N-module (Fig. EV4).

The ND4- and ND5-modules are important for the docking of other respiratory chain complexes. Inherited human CI deficiency associated with impaired assembly of very late CI subunits, such as NDUFA6 (N-module), showed that assemblies with complexes III and IV are still present in the patient cells, despite the lack of N-module (Alston et al, 2018). Interestingly, the connection to complexes III and IV does not occur with the incomplete CI membrane arm in RTN4IP1$^{KO}$ cells. This suggests that the necessary interfaces for supercomplex formation are indeed in the membrane arm, with NDUFA11 possibly playing an essential role at the interface with CIII (Calvaruso et al, 2012; Fang et al, 2021; Fernandez-Vizarra and Ugalde, 2022; Moreno-Lastres et al, 2012). Although the CI assembly defect in the *RTN4IP1* patient fibroblasts is not as dramatic as in RTN4IP1$^{KO}$ cells, there is a marked reduction in the preformed ND5-module and CI-containing supercomplex assembly when compared to controls (Figs. EV2 and EV3).

To better understand the complexity of how the preformed ND5-module attaches to complete the membrane arm, we examined the recently reported structure of CI at 2.39 Å resolution (PDB: 8OM1) (Grba et al, 2023). The ND5 subunit contains an exceptionally long lateral helix that extends over the ND4-module and into the ND2-module, suggesting that the late-stage assembly of the membrane arm requires a degree of structural flexibility to integrate this helix into the pre-assembled Q, ND1, ND2, and ND4 intermediate.

Complexome profiling revealed a clear absence or severe reduction of the subunits NDUFA10, NDUFA11, NDUFB4, and NDUFB2 in their respective intermediate modules (Fig. 4). NDUFB2, a distal subunit interacting with ND5, likely assembles at a late stage. In contrast, NDUFA10, part of the ND2-module, lacks direct interaction with the ND5 helix. Interestingly, NDUFA11, also within the ND2-module, makes direct contact with ND5 (Fig. 6A–C). These interactions are critical for the assembly and stability of CI. Notably, NDUFA11 serves as an interface subunit to CIII, suggesting that its absence could hinder supercomplex formation. NDUFB4 makes extended "clamp-like" contacts across ND5's surface with a more complex topology than NDUFA11. The loss of NDUFB4, may further impair late-stage membrane arm assembly. In summary, RTN4IP1 appears to play a crucial role in the late-stage assembly of the membrane arm, likely facilitating the integration of the ND5 helix into the preformed Q-, ND1-, ND2- and ND4-module intermediate and giving a signal to preformed N-module docking (Fig. 6D).

RTN4IP1 is the second example, along with PYURF, of a protein working to connect the interrelated pathways of CI assembly and CoQ biosynthesis. Interestingly, the only reported case of pathogenic human *PYURF* variants presented neonatally with profound metabolic acidosis and multisystem mitochondrial disease, which included optic atrophy (Rensvold et al, 2022), a similar clinical presentation to RTN4IP1 deficiency. The need for concerted regulation of both CI and CoQ biosynthesis could arise from different sources. First, it is plausible that such proteins are required at the interface of these two essential pathways to carefully tune the ratio of CoQ to CI for the specific redox requirements of OxPhos. Second, CoQ or intermediates in the biosynthetic pathway could be required as participants in the assembly process of CI. This hypothesis seems less well supported due to different reports of the activity and stability of CI in various mouse models of CoQ deficiency (Garcia-Corzo et al, 2013; Vazquez-Fonseca et al, 2019). Discriminating between these two possibilities would benefit from a rigorous assessment of assembled CI abundance in diverse models of CoQ deficiency. Although the mechanism by which PYURF coordinates CoQ biosynthesis and CI assembly is unclear, it is

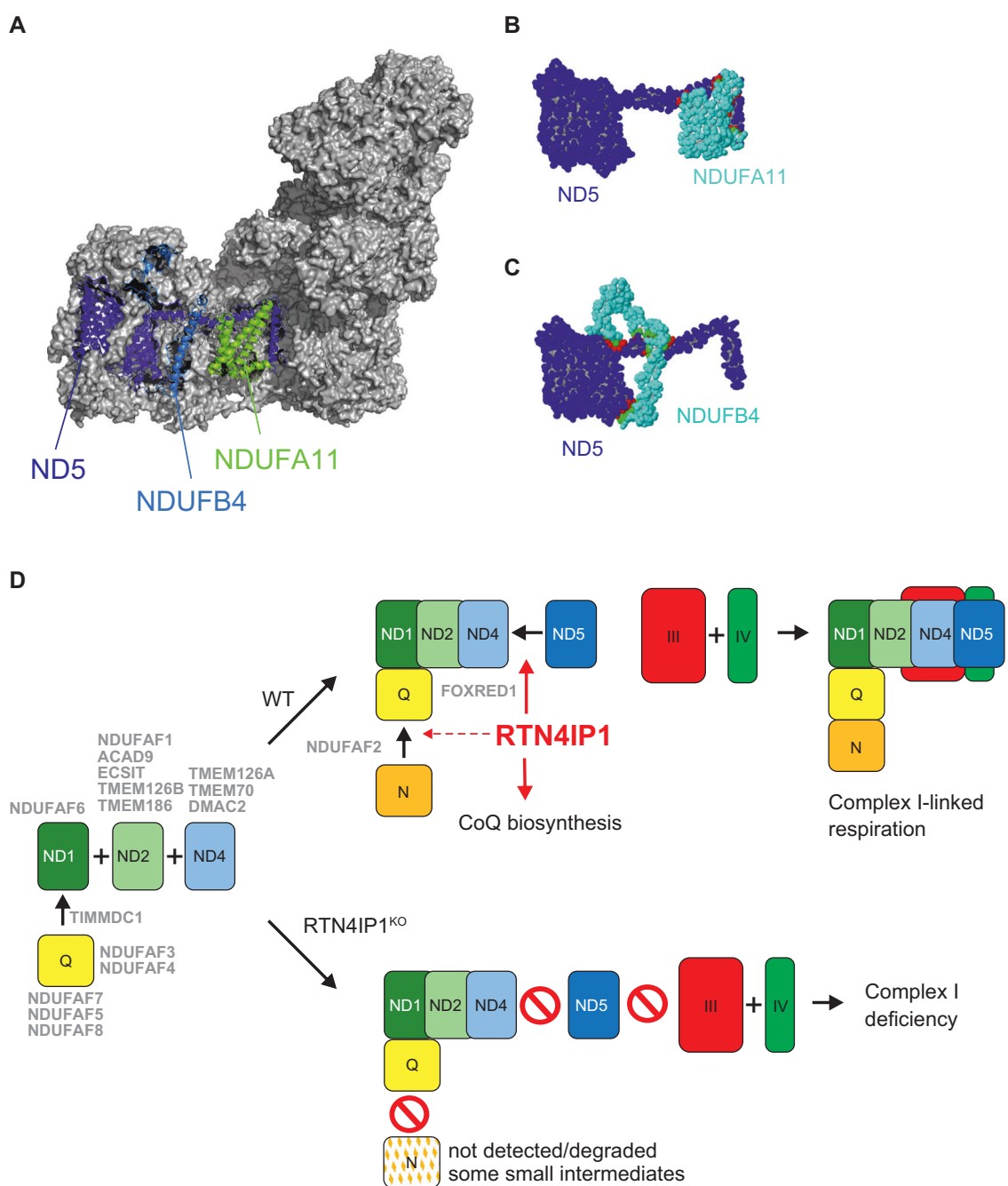

**Figure 6. Proposed role of RTN4IP1 in CI assembly.**

(A–C) Overview of mouse CI structure (PDB: 8OM1), highlighting the subunits of CI NDUFA11 (ND2-module), NDUFB4 (ND4-module), and ND5, which spans from the ND5-module through the ND4-module into the ND2-module. (B, C) Interface views showing the interaction between ND5 and NDUFA11 (B) or NDUFB4 (C), with the interface regions highlighted in red and light green, respectively. (D) Proposed model of RTN4IP1 function in CI assembly. Modules Q (yellow), ND1 (dark green), ND2 (light green), ND4 (light blue) and ND5 (dark blue) are pre-built using assembly factors (grey). Individual modules assemble under the control of additional assembly factors shown in the centre (grey). Under WT conditions (upper right), the membrane arm is complete, allowing the N-module (orange) to attach to form a fully active CI enzyme. Complexes III (red) and IV (green) are also added to fulfil the function of CI-dependent respiration as a supercomplex. The conditions under loss of RTN4IP1, i.e. in the RTN4IP1^KO cells, are shown on the lower right.

interesting to note that PYURF is reported to interact with COQ3, COQ5, and NDUFAF5, all methyltransferases in their respective pathways (Rensvold et al, 2022). Similarly, recent proximity interaction studies identify COQ3, COQ5, PYURF and NDUFAF7 as interacting partners of RTN4IP1 (Park et al, 2024). Curiously, NDUFAF7, a CI assembly factor, has been characterised as a protein methyltransferase that is required to methylate NDUFS2 in order to facilitate early CI assembly (Rhein et al, 2013). It is therefore tempting

to speculate that, like PYURF, RTN4IP1 enables the independent co-regulation of CoQ biosynthesis and CI assembly by facilitating critical methyltransferase reactions in both pathways.

Our protein-protein interaction assay showed an interaction between RTN4IP1 and NDUFV2 (Maio et al, 2017; Read et al, 2021) and NDUFV3 (Dibley et al, 2017; Yoval-Sanchez et al, 2022). NDUFV2 is positioned adjacent to the accessory CI flavoprotein NDUFV3 within the N-module. NDUFV2 is a core subunit of CI containing the N1a [$Fe_2$-$S_2$] cluster and is involved in electron transfer from NADH to ubiquinone. Interactions between RTN4IP1 and NDUFV2 suggest that RTN4IP1 may be required for early priming and maturation of the redox-sensitive N-module subunit of CI.

Additionally, while expression of RTN4IP1$^{WT}$, RTN4IP1$^{R103H}$ or RTN4IP1$^{G215A}$ in RTN4IP1$^{KO}$ cells fully rescued the CoQ biosynthetic defect (namely, oxidised $CoQ_{10}$ -representing the majority of the cellular CoQ pool and $PPHB_{10}$) (Fig. 5D), only a partial rescue of CI protein abundance and assembly was observed (Fig. 5A,C). This suggests that, although structurally CI has been restored (Fig. 5C), the functional impairment of CI is unable to fully reduce $CoQ_{10}$, resulting in a predominantly oxidised CoQ pool (Fig. 5D). Although, the localisation of WT and mutant RTN4IP1-FLAG constructs was not directly assessed, the full rescue of the CoQ defect and AEMS from isolated mitochondria support the correct mitochondrial targeting of the expressed constructs. These findings further support the potential function of RTN4IP1 in two distinct biological processes. Further analysis of a large-scale mitochondrial multi-omics study (Rensvold et al, 2022) allowed us to interrogate any reciprocal relationship between the CoQ and CI deficiencies (Fig. EV1A,B). Our findings support that these phenotypes arise from distinct biochemical functions of RTN4IP1. Nonetheless, testing additional catalytic RTN4IP1 mutants in future studies could help clarify whether NAD(P)H oxidoreductase activity of RTN4IP1 plays a direct mechanistic role in either or both pathways.

In summary, we have shown that the NAD(P)H oxidoreductase RTN4IP1 is required for normal CoQ production in humans. RTN4IP1 also functions as a late-stage assembly factor, crucial for the stability of the ND5-module and its docking to the ND4-module and maturation of the N-module subunits—processes which are prerequisites for forming a fully assembled and functional CI. RTN4IP1, therefore, represents a mitochondrial protein that appears to have independent roles in CI biogenesis and CoQ production.

# Methods

### Reagents and tools table

| Reagent/resource | Reference or source | Identifier or catalogue number |
| --- | --- | --- |
| **Experimental models** | | |
| *RTN4IP1* patient fibroblasts | (Charif et al, 2018) | Fig. 1 |
| U2OS wildtype | This study | Materials and Methods |
| U2OS RTN4IP1$^{KO+/-}$ | This study | Materials and Methods |
| U2OS RTN4IP1$^{KO-/-}$ | This study | Materials and Methods |

| Reagent/resource | Reference or source | Identifier or catalogue number |
| --- | --- | --- |
| Lenti-X 293T system | Takara | Cat# 632180 |
| **Recombinant DNA** | | |
| Expression plasmids | This study | Table EV1 |
| **Antibodies** | | |
| Anti-RTN4IP1 | Sigma | Cat# SAB1408126 |
| Anti-NDUFB8 | Abcam | Cat# ab110242 |
| Anti-SDHA | Abcam | Cat# ab14715 |
| Anti-UQCRC2 | Abcam | Cat# ab14745 |
| Anti-COXI | Abcam | Cat# ab14705 |
| Total OxPhos Human WB Antibody Cocktail | Abcam | Cat# ab110411 |
| Anti-NDUFA9 | Molecular Probes | Cat# A21344 |
| Anti-NDUFS1 | Proteintech | Cat# 12444-1-AP |
| Anti-NDUFA13 | Abcam | Cat# ab110240 |
| Anti-FLAG | Sigma | Cat# F1804 |
| Anti-beta-Actin | Abcam | Cat# ab8224 |
| Anti-VDAC1 | Abcam | Cat# ab14734 |
| Anti-GAPDH | Proteintech | Cat# 60004-1-Ig |
| Anti-rabbit HRP-conjugated secondary antibodies | Dako | Cat# P0399 |
| Anti-mouse HRP-conjugated secondary antibodies | Dako | Cat# P0260 |
| **Oligonucleotides and other sequence-based reagents** | | |
| PCR primers | This study | Table EV1 |
| **Chemicals, enzymes and other reagents** | | |
| SuperSignal West Pico PLUS Chemiluminescent Substrate | Thermo Fisher Scientific | Cat# 34577 |
| Polybrene | Sigma | Cat# TR-1003 |
| Puromycin | Thermo Fisher Scientific | Cat# A1113803 |
| Coenzyme $Q_8$ | Avanti Polar Lipids | Cat# 900151P-1mg |
| cOmplete Protease Inhibitor Cocktail | Sigma | Cat# 11697498001 |
| Benzonase | Sigma | Cat# E1014 |
| Trypsin | Promega | Cat# V5113 |
| Digitonin | Serva | Cat#19551.01 |
| ProteaseMAX surfactant | Promega | Cat# V2071 |
| DSSO | Thermo Fisher Scientific | Cat# A33433 |
| Dulbecco's modified Eagle's medium | Gibco™ | Cat# 31966047 |
| Foetal bovine serum | Gibco™ | Cat# 16140071 |
| Penicillin-streptomycin | Gibco™ | Cat# 15140122 |
| Non-essential amino acids | Gibco™ | Cat# 11140068 |
| Uridine | Sigma | Cat# U3003 |
| Cell Line Nucleofector® Kit V | Lonza | Cat# VCA-1003 |
| DirectPCR Lysis Reagent | Viagen | Cat# 102-T |

| Reagent/resource | Reference or source | Identifier or catalogue number |
|---|---|---|
| Proteinase K | Sigma | Cat# P4850 |
| Pyruvate | Sigma | Cat# P2256 |
| FCCP | Sigma | Cat# SML2959 |
| Antimycin A | Sigma | Cat# A8674 |
| Malate | Sigma | Cat# M1000 |
| Glutamate | Sigma | Cat# G1626 |
| ADP | Merck | Cat# 117105 |
| Rotenone | Sigma | Cat# R8875 |
| Succinate | Sigma | Cat# S2378 |
| Serva Blue G/Coomassie G250 | Serva | Cat# 35050.02 |
| n-dodecyl β-ᴅ-maltoside | MedChemExpress | Cat# HY-128974 |
| Novex® NativePAGE™ Bis-Tris Gel System | Thermo Fisher Scientific | Cat# BN1002BOX |
| **Software** | | |
| DatLab 6.1.0.7 | Oroboros Instruments | N/A |
| TraceFinder 5.1 | Thermo Fisher Scientific | N/A |
| Proteome Discoverer v3.2.0.450 | Thermo Fisher Scientific | N/A |
| MaxQuant v2.0.3.0 and v1.5.3.30 | (Cox and Mann, 2008) | N/A |
| Perseus v.1.6.15.0 | (Tyanova et al, 2016) | N/A |
| Prism v9.4.1 | GraphPad | N/A |
| **Other** | | |
| Q5 Site-Directed Mutagenesis Kit | NEB | Cat# E0554S |
| Lenti-X concentrator | Takara | Cat# 631231 |
| BCA Assay | Pierce | Cat# 23225 |
| Magnetic carboxylated SpeedBeads | Sigma | Cat# GE65152105050250 |
| Pierce peptide desalting columns | Thermo Fisher Scientific | Cat# 89851 |
| Magnetic anti-FLAG beads | Sigma | Cat# M8823-1ML |

## Ethical approval

Informed consent for diagnostic and research studies was obtained in accordance with the Declaration of Helsinki protocols and approved by the Newcastle and North Tyneside Local Research Ethics Committee (REC ref 2002/205).

## Mammalian cell culture

Primary age-matched control and *RTN4IP1* patient fibroblasts and CRISPR-Cas9-edited U2OS RTN4IP1^KO+/−, RTN4IP1^KO−/− and wild-type (WT) cell lines were grown in high glucose Dulbecco's modified Eagle's medium (DMEM, Gibco™ #31966047) supplemented with 10% foetal bovine serum (Gibco™ #16140071), 1%

penicillin-streptomycin (Gibco™ #15140122), 1x non-essential amino acids (Gibco™ #11140068) and 50 µg/mL uridine (Sigma #U3003) at 37 °C and 5% $CO_2$.

## Generation of RTN4IP1^KO cell models

CRISPR-Cas9 genome editing was used to generate RTN4IP1^KO and isogenic control cell models. Online CRISPR design tools were used to identify a 20 bp sequence single guide RNA (sgRNA) complementary to the target sequence in the *RTN4IP1* gene (GGATTAGTACTACCTCTCCT). The *RTN4IP1* sgRNA was cloned into the CRISPR-Cas9 expression vector px459 (Addgene). Wild-type U2OS cells were sub-cultured two days before nucleo-fection to achieve 80% confluency. Cells ($1 \times 10^6$) were nucleofected with the px459 expressing the 20 bp *RTN4IP1* sgRNA according to the manufacturer's instructions (Lonza #VCA-1003). Subsequently, cells expressing the px459 + *RTN4IP1* sgRNA plasmid were enriched by antibiotic selection for 48 h, and individual single clones were expanded for 2 weeks. Genomic DNA was isolated from expanded clones by incubating cell pellets in the presence of DirectPCR Lysis Reagent (Mouse Tail) (Viagen #102-T) and Proteinase K (Sigma #P4850) at 56 °C for 16 h, followed by 95 °C for 15 min. A target region containing the edited RTN4IP1 site has been PCR amplified (primers upon request) and Sanger sequenced. Isogenic wild-type control, heterozygous (c.112 C > T/+) and homozygous (c.111_127del/c.111_116delinsTCAA) RTN4IP1^KO cell lines were selected and used for subsequent experimental analysis.

## Western blot analysis

Cell lysis was performed on frozen cell pellets using buffer containing 50 mM Tris-HCl pH 7.5, 130 mM NaCl, 2 mM $MgCl_2$, 1 mM phenylmethylsulphonyl fluoride (PMSF), 1% Nonidet™ P-40 (v/v) and 1x EDTA-free protease inhibitor cocktail on ice for 20 min. Lysed samples were centrifuged at 1000×g for 5 min at 4 °C. Soluble protein fractions were retained and analysed by Western blotting. Equal amounts of protein extracts (minimum 40 µg) were electrophoretically separated on a 12% SDS polyacrylamide gel, followed by a wet transfer onto Immobilon-P polyvinylidene fluoride (PVDF) membrane (Merck Millipore #IPVH00005) using the Mini Trans-Blot Cell system (Bio-Rad).

## Immunoblotting

Primary and species appropriate HRP-conjugated secondary antibodies (Dako #P0260 and #P0399) were used; RTN4IP1 (Sigma #SAB1408126), NDUFB8 (Abcam #ab110242), SDHA (Abcam #ab14715), UQCRC2 (Abcam #ab14745), COXI (Abcam #ab14705), ATP5A (Abcam #ab14748), Total OxPhos Human WB Antibody Cocktail (Abcam #ab110411), NDUFA9 (Molecular Probes #A21344), NDUFS1 (Proteintech #12444-1-AP), NDUFA13 (Abcam #ab110240), FLAG (Sigma #F1804), Actin (Abcam #ab8224), VDAC1 (Abcam #ab14734) and GAPDH (Proteintech 60004-1-Ig). SuperSignal™ West Pico PLUS Chemiluminescent Substrate (Thermo Fisher Scientific #34577) and the ChemiDoc (Bio-Rad) system supported by the Image Lab software (Bio-Rad) were used for visualisation.

## Oxygen consumption

Mitochondrial basal respiration of intact cells was measured using high-resolution respirometry (Oxygraph-2k, Oroboros Instruments, Innsbruck, Austria) with DatLab software 6.1.0.7 (Oroboros Instruments, Innsbruck, Austria). Measurements were performed at 37 °C in full growth medium (DMEM, 15% FCS, pyruvate, uridine, non-essential amino acids, penicillin/streptomycin). Basal respiration (Basal) was measured for 20 min followed by titration of oligomycin (2.5 μM final concentration f.c.) to measure oligomycin-inhibited LEAK respiration. Subsequently, uncoupler FCCP (Sigma Aldrich, Munich, Germany) was titrated stepwise (0.5 μM per step, f.c.) until maximal uncoupled respiration (ETS) was reached. Residual oxygen consumption (ROX) was determined after sequential inhibition of CIII with antimycin A (2.5 μM f.c.) and CIV with KCN (2 mM f.c.). To analyse the effect of RTN4IP1 deletion on CI- and CII-linked respiration isolated mitochondria were used. CI- and CII-linked mitochondrial respiration was measured using high-resolution respirometry in MiR05-respiration media (110 mM sucrose, 60 mM potassium lactobionate, 0.5 mM EGTA, 3 mM $MgCl_2$, 20 mM taurine, 10 mM $KH_2PO_4$, 20 mM Hepes, 2 mg/ml bovine serum albumin, pH 7.1) at 37 °C. Mitochondria were added to the chamber followed by the addition of the following substrates and inhibitors (final concentration): 2 mM malate, 2.5 mM pyruvate, 10 mM glutamate, 2.5 mM ADP, 0.5 μM rotenone, 10 mM succinate and 2.5 μM antimycin A. Absolute respiration rates were corrected for ROX and normalised to the total number of cells per chamber, to the protein content or to citrate synthase activity.

## Generation of stable cell lines

Constructs of interest were cloned into pLVX-AcGFP1-N1 (Takara 632154) lentiviral vector under the CMV promoter. The RTN4IP1-Flag WT construct was amplified via PCR with Xba and XmaI restriction sites, digested, and ligated into the pre-digested pLVX-AvGFP1-N1 vector. Mutants were generated by site-directed mutagenesis (NEB E0554S) (Table EV1). Lentiviral particles were produced using the Lenti-X™ 293 T system (Takara 632180) with the pCMV-VSV-G (Addgene), pMDLg/pRRE (Addgene), and pRSV-Rev (Addgene) packaging plasmids. Viral media was collected 48 h post-transfection and processed with a Lenti-X concentrator (Takara 631231), then centrifuged (4 h, 600 × g, 4 °C) before aliquoting, flash freezing, and storing at −80 °C. To transduce cells, U2OS RTN4IP1$^{KO}$ cells were seeded in six-well plates, in the absence of penicillin-streptomycin, and incubated overnight. Media was supplemented with 0.5 μg/mL polybrene (Sigma #TR-1003), and 20 μl of lentivirus was added to each well. Cells were incubated for 48 h. The media was then exchanged for DMEM with 10% FBS and 0.5 μg/mL puromycin (Thermo Fisher Scientific, #A1113803) for 5 days. Cells were then passaged and expanded as normal and frozen down using freezing media (90% DMEM, 10% DMSO).

## Targeted CoQ measurement by LC–MS/MS

Determination of the CoQ content and redox state in mammalian cell culture was performed as previously described with modifications (Burger et al, 2020). In brief, frozen cell pellets were resuspended in 100 μL of PBS, and 5 μL was retained for a BCA assay to normalise lipidomic measurements to protein content. The

remaining cell resuspension was added to ice-cold extraction solution (600 μL acidified methanol [0.1% HCl final], 250 μL hexane, with 0.1 μM CoQ$_8$ internal standard [Avanti Polar Lipids]). Samples were vortexed and centrifuged (5 min, 17,000 × g, 4 °C), and the top hexane layer was retained. Extraction was repeated twice before the hexane layers were combined and dried under argon gas at room temperature. Extracted dried lipids were resuspended in methanol containing 2 mM ammonium formate and overlaid with argon.

LC–MS analysis was performed using a Thermo Vanquish Horizon UHPLC system coupled to a Thermo Exploris 240 Orbitrap mass spectrometer. For LC separation, a Vanquish binary pump system (Thermo Fisher Scientific) was used with a Waters Acquity CSH C18 column (100 mm × 2.1 mm, 1.7 μm particle size) held at 35 °C under a 300 μL/min flow rate. Mobile phase A consisted of 5 mM ammonium acetate in acetonitrile (ACN):$H_2O$ (70:30, v/v) with 125 μL/L acetic acid. Mobile phase B consisted of 5 mM ammonium acetate in isopropanol:ACN (90:10, v/v) with the same additive. For each sample run, mobile phase B was initially held at 2% for 2 min and then increased to 30% over 3 min. Mobile phase B was further increased to 50% over 1 min and 85% over 14 min, and then raised to 99% over 1 min and held for 4 min. The column was re-equilibrated for 5 min at 2% B before the next injection. Samples (5 μl) were injected by a Vanquish Split Sampler HT autosampler (Thermo Fisher Scientific), while the autosampler temperature was kept at 4 °C. The samples were ionised by a heated ESI source kept at a vaporiser temperature of 350 °C. Sheath gas was set to 50 units, auxiliary gas to 8 units, sweep gas to 1 unit, and the spray voltage was set to 3500 V for positive mode and 2500 V for negative mode. The inlet ion transfer tube temperature was kept at 325 °C with a 70% RF lens. For targeted analysis, the MS was operated separately in positive and negative mode, with targeted scans to oxidised CoQ$_{10}$ H$^+$ adduct (m/z 863.6912), oxidised CoQ$_{10}$ NH$_4^+$ adduct (m/z 880.7177), reduced CoQ$_{10}$H$_2$ H$^+$ adduct (m/z 865.7068), reduced CoQ$_{10}$H$_2$ NH$_4^+$ adduct (m/z 882.7334), oxidised CoQ$_8$ H$^+$ adduct (m/z 727.566), oxidised CoQ$_8$ NH$_4^+$ adduct (m/z 744.5935) and CoQ intermediate PPHB$_{10}$ H$^-$ adduct (m/z 817.6504). MS acquisition parameters include resolution of 45,000, HCD collision energy (45% for positive mode and 60% for negative mode), and 3 s dynamic exclusion. Automatic gain control targets were set to standard mode. The resulting CoQ intermediate data were processed using TraceFinder 5.1 (Thermo Fisher Scientific). Raw intensity values were normalised to the CoQ$_8$ internal standard and protein content as determined by BCA.

## LC–MS/MS proteomics

Cell culture pellets were resuspended in 2% SDS containing cOmplete Protease Inhibitor Cocktail (Sigma #11697498001) and heated at 95 °C for 5 min. Nucleic acids were sheared with benzonase (Sigma #E1014), and samples were incubated on ice for 15 min. Protein content was quantified using a BCA assay (Pierce 23225), and 100 μg of protein were alkylated and reduced in digestion solution (10 mM TCEP, 40 mM CAA, 100 mM Tris and pH 8.0) for 30 min at room temperature. Protein was subjected to single-pot, solid-phase-enhanced sample preparation (SP3) to remove detergent by incubating with magnetic carboxylated SpeedBeads (Sigma #GE65152105050250). After incubation (1 h), facilitating protein binding, beads were washed with 80% ethanol

and allowed to dry. The beads were resuspended in 100 mM Tris pH 8.0 and trypsin (Promega) was added to each sample in an estimated 50:1 protein:enzyme ratio for an overnight digest at 37 °C. The following day, the supernatant containing tryptic peptides was collected and acidified with TFA to a pH of 2.0. The peptides were desalted with Pierce peptide desalting spin columns (Thermo Fisher Scientific) and dried under vacuum (Thermo Fisher Scientific).

Samples were resuspended in 0.2% formic acid and subjected to LC–MS analysis. LC separation was performed using the Thermo Vanquish Neo UHPLC system. A 15 cm EASY-Spray PepMap Neo UHPLC C18 column (150 mm × 75 μm, 3 μm) was used with a 24 min gradient using mobile phase A consisting of 0.1% formic acid in $H_2O$, and mobile phase B consisting of 0.1% formic acid in ACN:$H_2O$ (80:20, v/v). An EASY-Spray source was used, and the temperature was set at 35 °C. Each sample was first held at 4% B for 0.5 min with an 800 nL/min flow rate, then the flow rate was dropped to 500 nL/min and increased to 8% B over 0.1 min and held there for 0.3 min. The gradient was then increased to 22.5% B over 13.9 min, to 35% B over 6.9 min, and 55% B over 0.4 min. The flow rate was adjusted to 800 nL/min and increased to 99% B over 0.5 min, followed by 2.3 min at 99% for column washing. An Acclaim PepMap C18 HPLC trap column (20 mm × 75 μm, 3 μm) was used for sample loading.

MS detection was performed with a Thermo Astral mass spectrometer in positive mode. The source voltage was 1.9 kV, the ion transfer tube temperature was set to 280 °C, and RF lens was at 40%. Full MS spectra were acquired from m/z 380 to 980 at the Orbitrap resolution of 240,000, with the normalised AGC target of 500% (5E6) and a maximum injection time of 3 ms. Data-independent acquisition (DIA) was performed with the Astral analyser with 300 isolation windows of 2-Th scanning from 380 to 980 m/z. The isolated ions were fragmented using HCD with 25% normalised collision energy. Other settings for DIA include an RF lens of 40%, a normalised ACG target of 500% (5E4), and a maximum injection time of 3 ms.

Raw files were analysed by the CHIMERYS search algorithm with INFERYS Rescoring incorporated in Proteome Discoverer v.3.2.0.450 software against human databases downloaded from Uniprot. MS2 Apex (quan in all files) was used as the quantification method for the searches.

## Blue-native electrophoresis and in-gel activity stains

Sample preparation and blue-native electrophoresis (BNE) of cultured cell pellets were essentially done as previously described (Oláhová et al, 2015; Wittig et al, 2006). Briefly: For BNE cells were collected by scraping and centrifugation, further disrupted using a precooled motor-driven glass/Teflon Potter-Elvehjem homogeniser at 2000 rpm and 40 strokes. Homogenates were centrifuged for 15 min at 600 × *g* to remove nuclei, cell debris, and intact cells. Mitochondrial membranes were sedimented by centrifugation of the supernatant for 15 min at 22,000 × *g*. Mitochondria-enriched membranes from 20 mg cells were resuspended in 35 μl solubilisation buffer (50 mM imidazole pH 7, 50 mM NaCl, 1 mM EDTA and 2 mM aminocaproic acid) and solubilized with 10 μl 20% digitonin (Serva). Samples were supplemented with 2.5 μl 5% Coomassie G250 in 500 mM aminocaproic acid and 5 μl 0.1% Ponceau S in 50% glycerol. Equal protein amounts of samples were loaded onto a 3 to 18% polyacrylamide gradient gel

(dimension 14 × 14 cm). After native electrophoresis in a cold chamber, blue-native gels were fixed in 50% (v/v) methanol, 10% (v/v) acetic acid, 10 mM ammonium acetate for 30 min and stained with Coomassie (0.025% Serva Blue G, 10% (v/v) acetic acid). BNE and immunoblotting analysis of *n*-dodecyl β-D-maltoside (DDM) solubilised mitochondria isolated from U2OS cells was performed as previously described (Oláhová et al, 2015). Briefly, mitochondrial membranes were solubilised using 2 mg/mg protein DDM on ice for 20 min and samples 100 μg were separated on a 4 to 16% native polyacrylamide gel following manufacturers' instructions (Novex® NativePAGE™ Bis-Tris Gel System and reagents). Separated OxPhos complexes were transferred to a PVDF membrane and immunoblotted for CI, CII and CV. In-gel activity stains were performed as described in (Wittig et al, 2007).

## Complexome profiling

Enriched mitochondrial proteins were extracted from U2OS cells and patient fibroblasts. Mitochondrial membranes were solubilised with digitonin as described in (Giese et al, 2021). Equal amounts of solubilised mitochondrial protein extracts were subjected to a 3 to 18% polyacrylamide gradient gel (14 cm × 14 cm) for BN-PAGE as outlined in (Wittig et al, 2006). The gel was then stained with Coomassie blue and cut into equal fractions, then transferred to 96-well filter plates. The gel fractions were then destained in 50 mM ammonium bicarbonate (ABC), followed by protein reduction using 10 mM DTT and alkylation in 20 mM iodoacetamide. Protein digestion was carried out in digestion solution (5 ng trypsin/μl in 50 mM ABC, 10% ACN, 0.01% (m/v) ProteaseMAX surfactant (Promega), 1 mM $CaCl_2$) at 37 °C for at least 12 h. The peptides were dried in a SpeedVac (Thermo Fisher Scientific) and eluted in 1% ACN and 0.5% formic acid into a new 96-well plate. Nano liquid chromatography and mass spectrometry (nanoLC/MS) was carried out on Thermo Scientific™ Q Exactive Plus equipped with an ultra-high performance liquid chromatography unit Dionex Ultimate 3000 (Thermo Fisher) and Nanospray Flex Ion-Source (Thermo Fisher Scientific). The data was analysed using MaxQuant software at default settings, and the recorded intensity-based absolute quantifications (iBAQ) values were normalised to control cell membranes.

## Affinity enrichment cross-linking mass spectrometry

Crude mitochondria were isolated from cells as described above and subjected to chemical cross-linking (0.5 mM DSSO (Thermo Fisher Scientific #A33433), 1 h, RT). Cross-linking was quenched with 100 mM Tris, pH 8.0 followed by centrifugation (15,000 × *g*, 5 min, 4 °C). The mitochondria were then solubilised with 50 mM imidazole, 500 mM 6-hexaminocaproic acid, 1 mM EDTA and 6 g/g digitonin. The bait protein and cross-linked interactors were then enriched by FLAG immunoprecipitation (IP) using magnetic anti-FLAG beads (Sigma #M8823-1ML), washed and subjected to on-bead tryptic digest. The on-bead cross-linked proteins were denatured with 2 M urea in 200 mM Tris, pH 8.0, then reduced with 5 mM DTT for 30 min at 56 °C and alkylated with 15 mM iodoacetamide for 30 min at RT in the dark. The proteins on-bead were digested overnight at RT with 1 μg trypsin (Promega V5113). The digested supernatant was acidified with 10% TFA to a pH of 2 and desalted with Pierce peptide desalting spin columns (Thermo

Fisher Scientific), then dried under vacuum using a Speed Vac (Thermo Scientific) and stored at $-80\,°C$ until MS analysis. Samples were resuspended in 0.2% formic acid and subjected to LC–MS analysis with a Thermo Vanquish Neo UHPLC system coupled to a Thermo Astral mass spectrometer, as described above. Raw files were searched with Proteome Discoverer v.3.2.0.450 software, as described above. The resulting data were analysed by Perseus v.1.6.15.0 software.

## Data availability

The proteomics and lipidomics data have been deposited to MassIVE repository (https://massive.ucsd.edu/ProteoSAFe/static/massive.jsp) with the following link and study ID: MSV000098108. The mass spectrometry complexomics data together with a detailed description have been deposited to the ProteomeXchange Consortium via the PRIDE partner repository (Perez-Riverol et al, 2022) with the following dataset identifiers: PXD055511 and PXD064861.

The source data of this paper are collected in the following database record: biostudies:S-SCDT-10_1038-S44318-025-00533-x.

## Peer review information

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

## Acknowledgements

This work was supported by the Wellcome Centre for Mitochondrial Research (203105/Z/16/Z) (RNL, ZMAC-L and RWT); Biochemical Society Vacation Scholarship funding (MO); a Barbour Foundation PhD studentship awarded to JJC; grants from the Deutsche Forschungsgemeinschaft (DFG), FOR5046 grant number WI-3728/1-1 (project number 426950122), WI-3728/3-1 (project number 515944830), SFB1531 (project S01#456687919) and the German Federal Ministry of Education and Research (BMBF, Bonn, Germany) grant to the German Network for Mitochondrial Disorders (mitoNET, 01GM1906D) (IW); NIH grants R01NS112381, R01NS131322 (AG); and the NIH award R35GM131795 and funds from the BJC Investigator Programme (DJP). MO and RWT receive additional financial support from the Pathology Society and Mito Foundation. MO is supported by Fight for Sight and the Academy of Medical Sciences. RWT is also supported by the UK NIHR Biomedical Research Centre in Age and Age-Related Diseases award to the Newcastle upon Tyne Hospitals NHS Foundation, the Lily Foundation, LifeArc and the UK NHS Highly Specialised Service for Rare Mitochondrial Disorders. We thank Jana Meisterknecht for excellent technical assistance.

## Author contributions

**Monika Oláhová**: Conceptualisation; Supervision; Funding acquisition; Data curation, Formal analysis, Investigation; Writing—original draft; Writing—review and editing. **Rachel M Guerra**: Data curation; Formal analysis; Investigation; Writing—original draft; Writing—review and editing. **Jack J Collier**: Data curation; Formal analysis; Investigation; Writing—original draft. **Juliana Heidler**: Data curation; Formal analysis; Investigation. **Kyle Thompson**: Data curation; Formal analysis; Investigation. **Chelsea R White**: Data curation; Formal analysis; Investigation. **Paulina Castañeda-Tamez**: Data curation; Formal analysis; Investigation. **Alfredo Cabrera-Orefice**: Data curation; Formal analysis; Investigation. **Robert N Lightowlers**: Supervision; Writing—review and editing. **Zofia M A Chrzanowska-Lightowlers**: Supervision; Writing—review and editing. **Alexander Galkin**: Data curation; Formal analysis; Investigation; Writing—review and editing. **Ilka Wittig**: Conceptualisation; Supervision; Funding acquisition; Writing—original draft; Writing—review and editing. **David J Pagliarini**: Conceptualisation; Supervision; Funding acquisition; Writing—original draft; Writing—review and editing. **Robert W Taylor**: Conceptualisation; Supervision; Funding acquisition; Writing—original draft; Project administration; Writing—review and editing.

Source data underlying figure panels in this paper may have individual authorship assigned. Where available, figure panel/source data authorship is listed in the following database record: biostudies:S-SCDT-10_1038-S44318-025-00533-x.

## Disclosure and competing interests statement

The authors declare no competing interests.

# Expanded View Figures

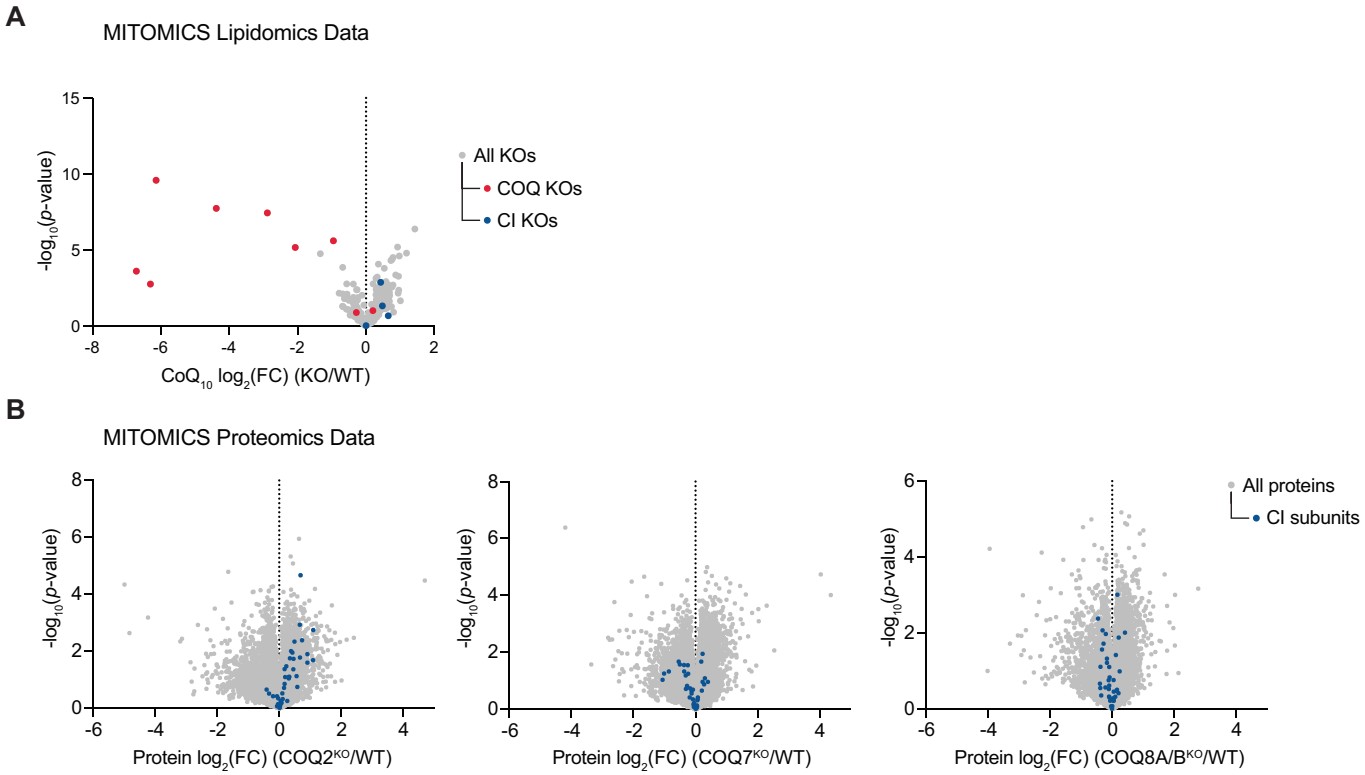

**A** MITOMICS Lipidomics Data

**B** MITOMICS Proteomics Data

**Figure EV1. Perturbations in CI and COQ proteins do not lead to interdependent phenotypes.**

(**A, B**) Targeted lipidomics for $CoQ_{10}$ abundance (**A**) and proteomics (**B**) analyses from the MITOMICS study (Rensvold et al, 2022) investigating the reciprocal interaction between COQ and CI proteins. Volcano plots depict the $\log_2$ fold change of $CoQ_{10}$ (**A**) or protein (**B**) abundance of all HAP1 knock out cell lines relative to WT. In (**A**), COQ KO cell lines are highlighted in red, CI KO cell lines are highlighted in blue. In (**B**), CI subunits are highlighted in blue. Data shown as mean ($n = 3$), two-sided Student's $t$-test.

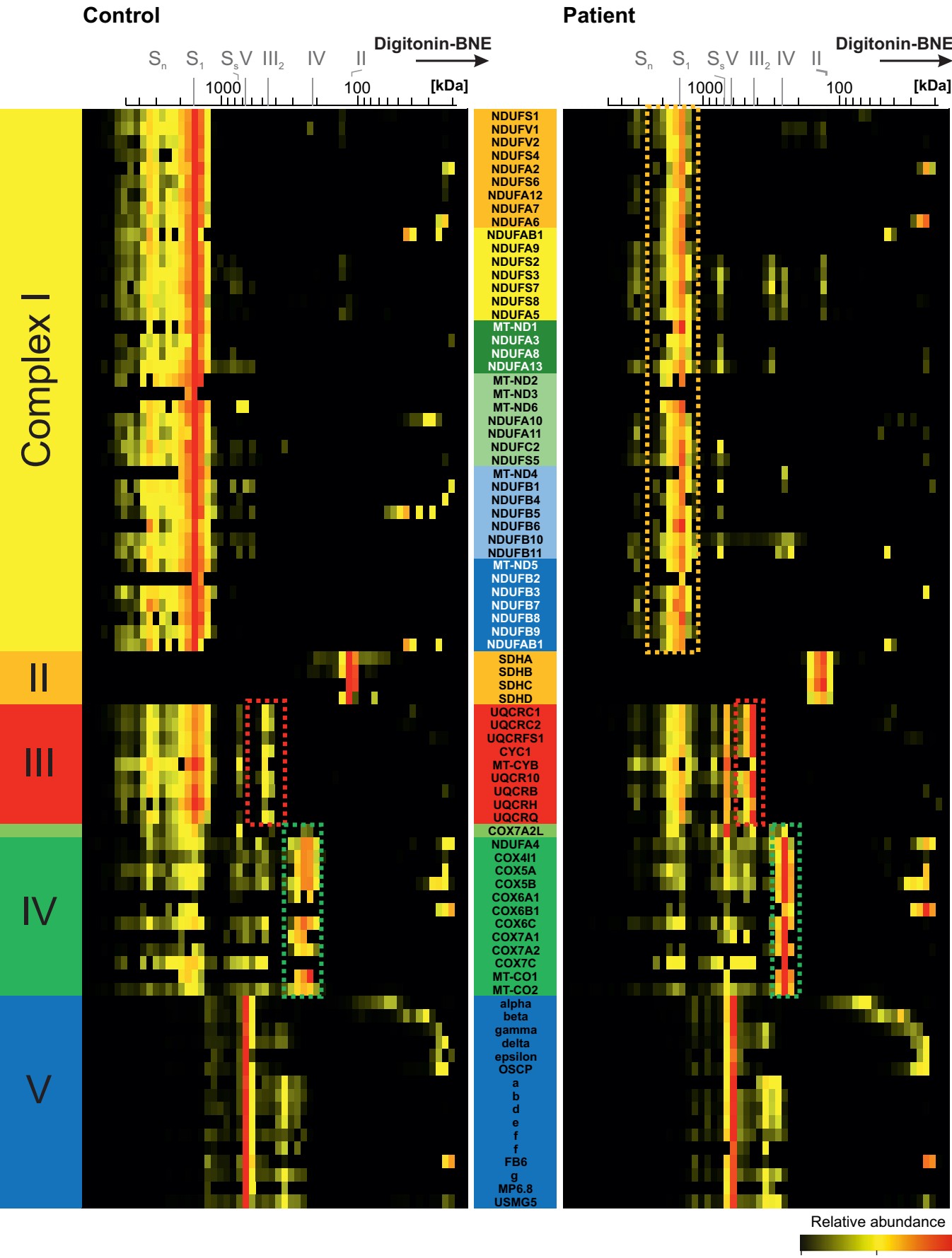

**Figure EV2. OxPhos complexes of *RTN4IP1* patient and control fibroblasts analysed by complexome profiling.**

Complexome profiling data were presented as a heatmap, corresponding to subunits of individual OxPhos complexes I-V. Mitochondrial complexes were solubilised with digitonin, separated on blue-native gels (BNE), cut into fractions and analysed by quantitative mass spectrometry. Assignment of complexes: complex II (II); complex III dimer (III$_2$); complex IV (IV); complex V (V); small supercomplex of CIII$_2$ and CIV (S$_s$); supercomplex containing CI, III$_2$ and 1 copy of CIV (S$_1$) and higher order supercomlexes (S$_n$). The relative abundance of each protein was represented from low to high according to the colour scale illustrated on the bottom right.

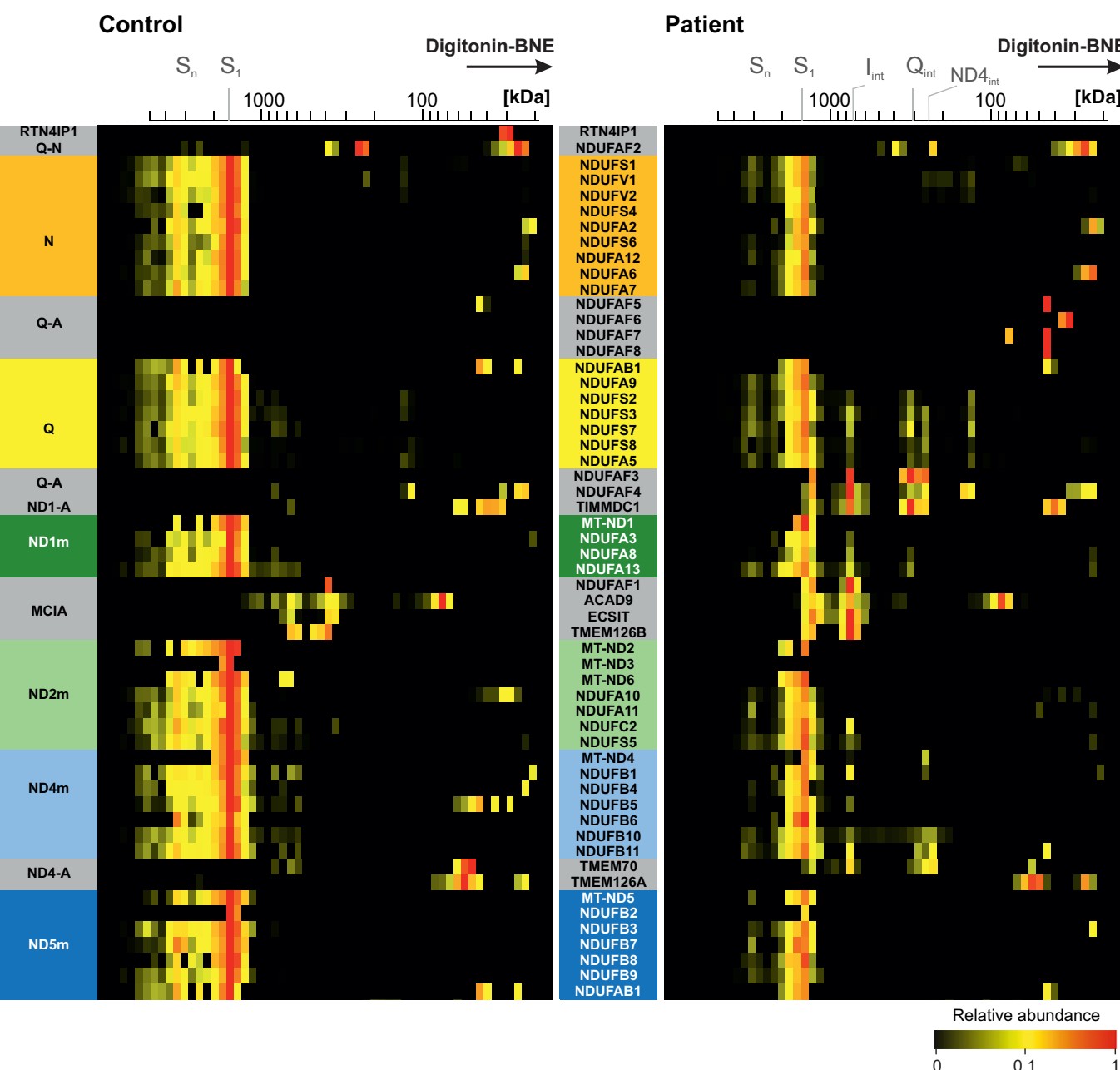

**Figure EV3.   CI assembly in *RTN4IP1*-derived patient fibroblasts corresponding to control cells.**

Complexome profiling data of CI and identified assembly factors were sorted to their assembly modules and presented as a heatmap. Assignment of complexes: higher order supercomplexes ($S_n$); supercomplex $S_1$ of CI, CIII$_2$ and CIV ($S_1$); intermediate of CI including modules Q, ND1m, ND2m, ND4m and assembly factors ($I_{int}$); intermediate of Q-module containing assembly factors ($Q_{int}$) and intermediate of ND4-module containing assembly factors ($ND4_{int}$).

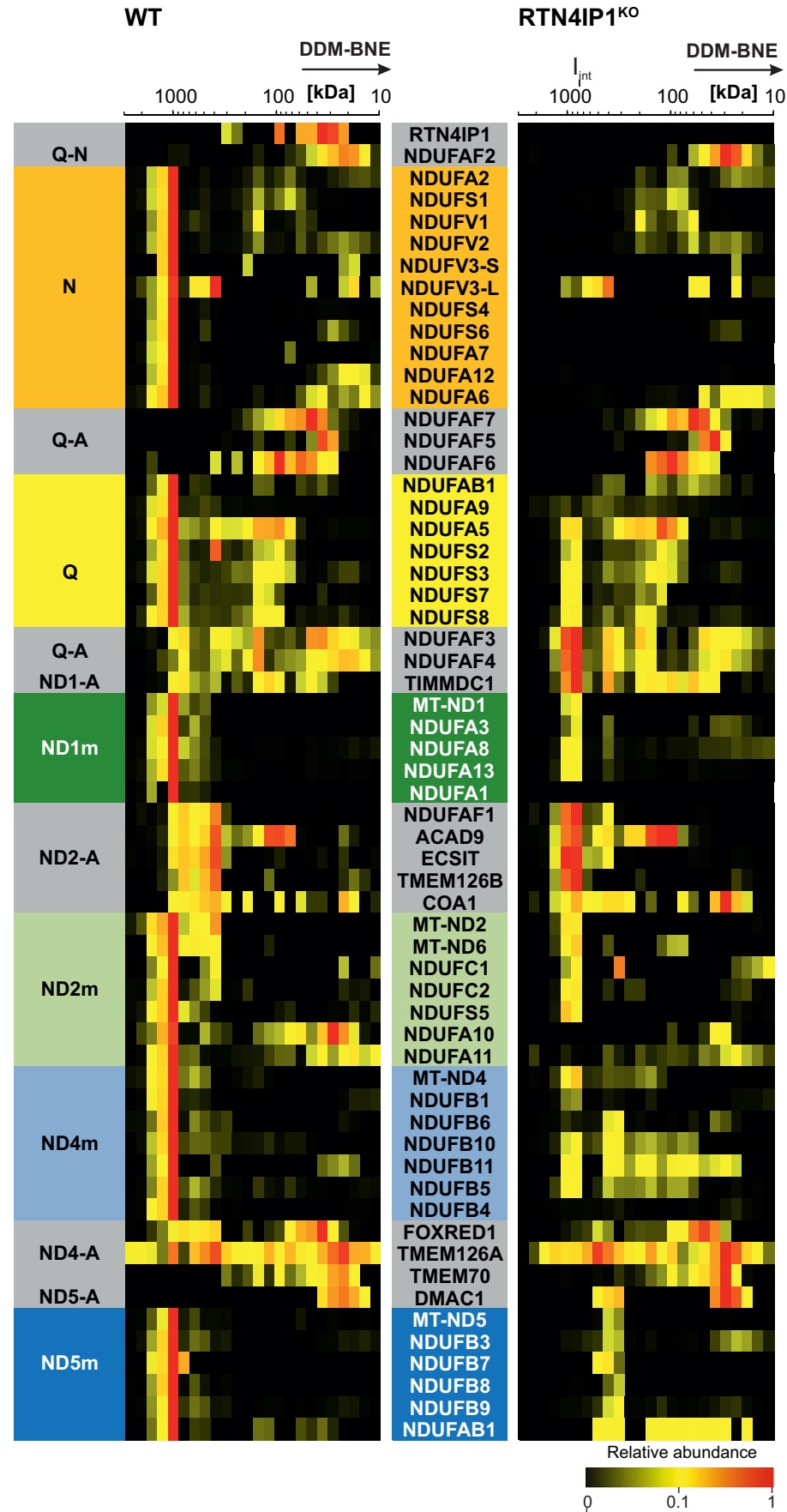

**Figure EV4. CI assembly in U2OS WT and U2OS RTN4IP1[KO] cells.**

DDM-solubilised mitochondria complexome profiling data of CI and identified assembly factors were sorted to their assembly modules and presented as heatmap. Assignment of complexes: intermediate of CI including modules Q, ND1m, ND2m, ND4m and assembly factors ($I_{int}$).

**A**

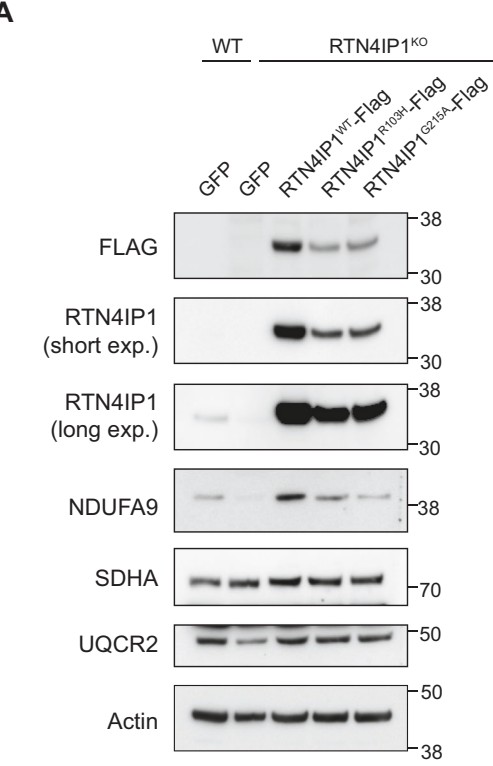

**B**

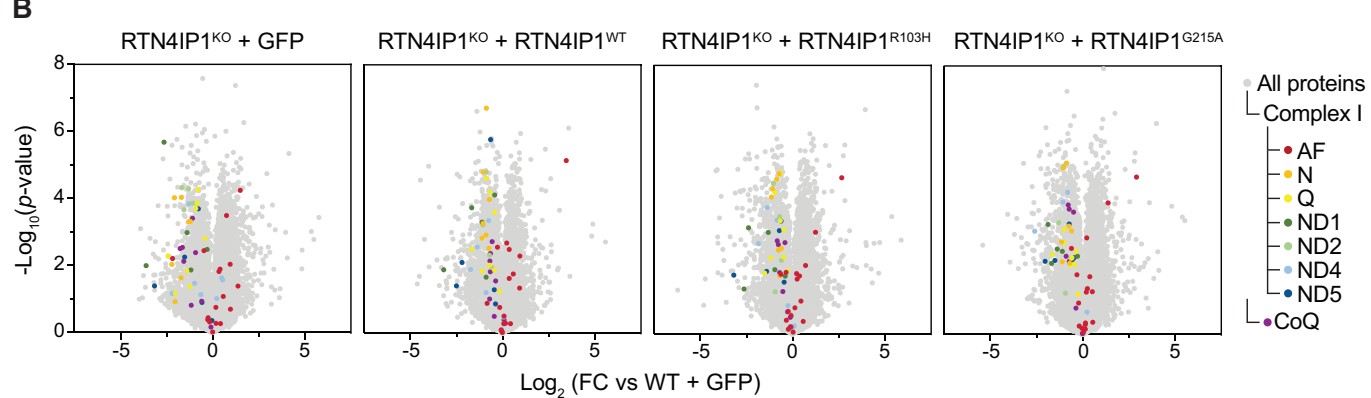

-Log$_{10}$(*p*-value)

Log$_2$ (FC vs WT + GFP)

All proteins
Complex I
- AF
- N
- Q
- ND1
- ND2
- ND4
- ND5
CoQ

RTN4IP1$^{KO}$ + GFP   RTN4IP1$^{KO}$ + RTN4IP1$^{WT}$   RTN4IP1$^{KO}$ + RTN4IP1$^{R103H}$   RTN4IP1$^{KO}$ + RTN4IP1$^{G215A}$

**C**

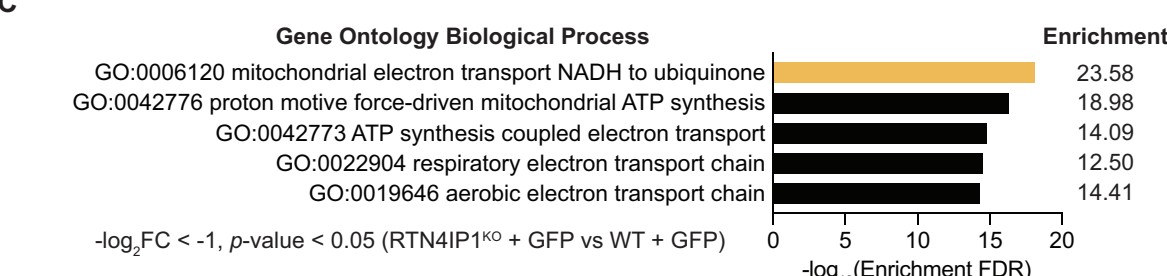

| Gene Ontology Biological Process | Enrichment |
|---|---|
| GO:0006120 mitochondrial electron transport NADH to ubiquinone | 23.58 |
| GO:0042776 proton motive force-driven mitochondrial ATP synthesis | 18.98 |
| GO:0042773 ATP synthesis coupled electron transport | 14.09 |
| GO:0022904 respiratory electron transport chain | 12.50 |
| GO:0019646 aerobic electron transport chain | 14.41 |

-log$_2$FC < -1, *p*-value < 0.05 (RTN4IP1$^{KO}$ + GFP vs WT + GFP)

-log$_{10}$(Enrichment FDR)

◀  **Figure EV5.  Validation of the U2OS RTN4IP1 rescue cell lines.**

(A) Western blotting analysis validating the stable expression of RTN4IP1-FLAG constructs in the U2OS RTN4IP1$^{KO}$ cell lines. WT and RTN4IP1$^{KO}$ cells were infected with a GFP or FLAG-tagged RTN4IP1-encoding lentivirus, resulting in the generation of WT and RTN4IP1$^{KO}$ cells expressing GFP, and RTN4IP1$^{KO}$ cells expressing RTN4IP1-FLAG mutants (RTN4IP1$^{WT}$, RTN4IP1$^{R103H}$ or RTN4IP1$^{G215A}$). Actin and SDHA were used as loading controls. (B) Volcano plot of proteomics experiments depicted in main Fig. 5A,B, showing protein abundances in U2OS RTN4IP1$^{KO}$(+*GFP*) or U2OS RTN4IP1$^{KO}$(+RTN4IP$^{WT}$, RTN4IP1$^{R103H}$ or RTN4IP1$^{G215A}$) cells relative to WT(+*GFP*). CI proteins are highlighted, and colour coordinated to match individual CI modules. Data shown as mean ($n = 3$), two-sided Student's *t*-test. (C) Gene Ontology of Biological Process showing enrichment of mitochondrial electron transport from NADH to ubiquinone proteins based on proteomics analysis of RTN4IP1$^{KO}$(+*GFP*) versus WT(+*GFP*), for proteins whose log$_2$ FC $<-1$ and *p* value $< 0.05$. FDR for the relevant gene sets (top to bottom): 7.58e-19, 4.81e-17, 1.64e-15, 3.08e-15, 4.82e-15; hypergeometric test. Source data are available online for this figure.

