## [Peer Review File · The EMBO Journal]

RTN4IP1 is required for the final stages of mitochondrial complex I assembly and CoQ biosynthesis

Monika Oláhová, Rachel Guerra, Jack Collier, Juliana Heidler, Kyle Thompson, Chelsea White, Paulina Castañeda-Tamez, Alfredo Cabrera-Orefice, Robert Lightowlers, Zofia Chrzanowska-Lightowlers, Alexander Galkin, Ilka Wittig, David Pagliarini, and Robert Taylor

Corresponding author(s): Robert Taylor (Robert.Taylor@newcastle.ac.uk) , Monika Oláhová (monika.winter@northumbria.ac.uk)

Review Timeline:

Submission Date:	3rd Nov 24
Editorial Decision:	20th Dec 24
Revision Received:	11th Jun 25
Editorial Decision:	30th Jun 25
Revision Received:	10th Jul 25
Accepted:	22nd Jul 25

Editor: William Teale

Transaction Report:

Dear Rob,

Thank you again for the submission of your manuscript entitled "RTN4IP1 is essential for the final stages of mitochondrial complex I assembly and CoQ biosynthesis" and for your patience during the review process. We have now received the reports from the referees, which I copy below.

As you can see from their comments, while referee #3 expressed enthusiasm for the scope and execution of the study, referees #2 and #1 both expressed concerns over the concrete mechanistic insights gained. At this stage, I am considering this manuscript as a better fit for EMBO Reports; however, it would certainly be a good idea to discuss this on a Zoom call in the new year. In the meantime, I am happy to formally invite you to address the comments of all referees in a revised version of the manuscript. That said, I must emphasise that, depending on the revisions that are feasible within a timeframe of three to six months, I may recommend publication in EMBO Reports after re-review.

I should add that it is The EMBO Journal policy to allow only a single major round of revision and that it is therefore important to resolve the main concerns at this stage. Please contact me if you have any questions, need further input on the referee comments or if you anticipate any problems in addressing any of their points. Please, follow the instructions below when preparing your manuscript for resubmission.

I would also like to point out that as a matter of policy, competing manuscripts published during this period will not be taken into consideration in our assessment of the novelty presented by your study ("scooping" protection). We have extended this 'scooping protection policy' beyond the usual 3 month revision timeline to cover the period required for a full revision to address the essential experimental issues. Please contact me if you see a paper with related content published elsewhere to discuss the appropriate course of action.

Again, please contact me at any time during revision if you need any help or have further questions.

Thank you very much again for the opportunity to consider your work for publication. I look forward to your revision.

Best regards,

William

William Teale, Ph.D.
Editor
The EMBO Journal

When submitting your revised manuscript, please carefully review the instructions below and include the following items:

- 1) a .docx formatted version of the manuscript text (including legends for main figures, EV figures and tables). Please make sure that the changes are highlighted to be clearly visible.
- 2) individual production quality figure files as .eps, .tif, .jpg (one file per figure).
- 3) a .docx formatted letter INCLUDING the reviewers' reports and your detailed point-by-point response to their comments. As part of the EMBO Press transparent editorial process, the point-by-point response is part of the Review Process File (RPF), which will be published alongside your paper.
- 4) a complete author checklist, which you can download from our author guidelines ([https://wol-prod-cdn.literatumonline.com/pb-assets/embo-site/Author Checklist%20-%20EMBO%20J-1561436015657.xlsx](https://wol-prod-cdn.literatumonline.com/pb-assets/embo-site/Author%20Checklist%20-%20EMBO%20J-1561436015657.xlsx)). Please insert information in the checklist that is also reflected in the manuscript. The completed author checklist will also be part of the RPF.
- 5) Please note that all corresponding authors are required to supply an ORCID ID for their name upon submission of a revised manuscript.
- 6) We require a 'Data Availability' section after the Materials and Methods. Before submitting your revision, primary datasets produced in this study need to be deposited in an appropriate public database, and the accession numbers and database listed

under 'Data Availability'. Please remember to provide a reviewer password if the datasets are not yet public (see <https://www.embopress.org/page/journal/14602075/authorguide#datadeposition>). If no data deposition in external databases is needed for this paper, please then state in this section: This study includes no data deposited in external repositories. Note that the Data Availability Section is restricted to new primary data that are part of this study.

Note - All links should resolve to a page where the data can be accessed.

8) For data quantification: please specify the name of the statistical test used to generate error bars and P values, the number (n) of independent experiments (specify technical or biological replicates) underlying each data point and the test used to calculate p-values in each figure legend. The figure legends should contain a basic description of n, P and the test applied. Graphs must include a description of the bars and the error bars (s.d., s.e.m.).

9) We would also encourage you to include the source data for figure panels that show essential data. Numerical data can be provided as individual .xls or .csv files (including a tab describing the data). For 'blots' or microscopy, uncropped images should be submitted (using a zip archive or a single pdf per main figure if multiple images need to be supplied for one panel). Additional information on source data and instruction on how to label the files are available at .

10) We replaced Supplementary Information with Expanded View (EV) Figures and Tables that are collapsible/expandable online (see examples in <https://www.embopress.org/doi/10.15252/embj.201695874>). A maximum of 5 EV Figures can be typeset. EV Figures should be cited as 'Figure EV1, Figure EV2" etc. in the text and their respective legends should be included in the main text after the legends of regular figures.

12) Our journal encourages inclusion of *data citations in the reference list* to directly cite datasets that were re-used and obtained from public databases. Data citations in the article text are distinct from normal bibliographical citations and should directly link to the database records from which the data can be accessed. In the main text, data citations are formatted as follows: "Data ref: Smith et al, 2001" or "Data ref: NCBI Sequence Read Archive PRJNA342805, 2017". In the Reference list, data citations must be labeled with "[DATASET]". A data reference must provide the database name, accession number/identifiers and a resolvable link to the landing page from which the data can be accessed at the end of the reference. Further instructions are available at .

13) In order to increase the reproducibility and reach of your work, The EMBO Journal includes a table of reagents that were used in the study. Please provide this along with your revisions.

Further instructions for preparing your revised manuscript:

We realize that it is difficult to revise to a specific deadline. In the interest of protecting the conceptual advance provided by the work, we recommend a revision within 3 months (20th Mar 2025). Please discuss the revision progress ahead of this time with the editor if you require more time to complete the revisions. Use the link below to submit your revision:

Referee #1:

Built from 45 known subunits coming from two genomes and containing 7 iron sulfur clusters, respiratory complex I assembly is a complicated modular process that is assisted by at minimum 18 known assembly components. Cast in this light, it is unsurprising that complex I deficiencies account for an estimated ~30% of cases of primary mitochondrial disease. It was previously shown that disease-associated RTN4IP1 variants result in isolated complex I deficiency. At the time of this initial finding, RTN4IP1 was classified as a mitochondrial protein of unknown function although it was identified in a CRISPR screen as important for oxidative phosphorylation. Here, Oláhová et al confirm that complex I expression and assembly is disrupted in RTN4IP1 variant patient-derived fibroblasts and newly generated RTN4IP1-knockout U2OS cells and formally establish that this decrease in expression and assembly results in a significant drop in complex I-linked respiration. Using complexome profiling, evidence is provided that RTN4IP1 is a complex I assembly factor that operates at a late stage of assembly, suggested to reflect a defect in the engagement of the ND5 and ND4 modules which results in a decreased abundance of N module components that fail to assemble in high molecular weight supercomplexes. Based on a recent paper that established that RTN4IP1 is a matrix-localized NADPH oxidoreductase that promotes the activity of COQ3, CoQ levels were shown here to be decreased in patient-derived fibroblasts and RTN4IP1-KO cells which also accumulated the CoQ precursor, PPHB10. While the results largely support the conclusions being made, they are, with the exception of the complexome profiling, confirmatory in nature and lacking in mechanistic insight. For example, while the complexome profiling does clearly indicate that complex I assembly is impaired, likely at the step of ND5 module docking onto ND4, the fact that RNT4IP1 does not co-migrate in an informative manner leaves it very much unclear as to how it contributes to this process. And while the twin roles in complex I assembly and CoQ biosynthesis are intriguing, no effort has been made to determine if these processes are linked or separate.

Major Points

1. Some effort to determine mechanistically how RTN4IP1 promotes complex I assembly is needed. The current discussion section nicely outlines potential productive avenues to pursue.
2. In surveying the literature, I did not notice if it was known as to whether defects in complex I assembly impact CoQ biosynthesis and/or vice versa. Is CoQ biosynthesis perturbed in the absence of other complex I assembly factors? Is complex I assembly impaired when CoQ biosynthesis is disrupted in general?
3. Is RTN4IP1 activity required for complex I assembly and/or CoQ biosynthesis?

Minor Points

1. It is argued that the ND5 module was "not able to join to the ND4-module and the large Q/ND1/ND2-intermediate." However, the ND5 module does appear to co-migrate with some proportion of the ND4 module bounded by the dashed white box. While

ND5 module assembly into the S and lint complexes is clearly disrupted in the absence of RTN4IP1, I am not convinced that it is totally unable to associate with a ND4 module-containing assembly intermediate.

2. The attention on the absence of FOXRED1 seems arbitrary given that DMAC2 assembly is also notably altered in absence of RTN4IP1.

3. For patient fibroblasts (EV4), it seems as if TMEM126A is missing from lint and FOXRED1 and DMAC2 not detected. Is this difference in TMEM126A between KO and patient cells informative?

4. In comparison to the RTN4IP1-KO cells, only half of the ND1 module co-migrates with the CI intermediate (lint) in WT extracts. Given this seemingly major difference, it is unclear why the presence of this intermediate in WT extracts is noteworthy. Any ideas as to what this different pattern may reflect?

5. The labels in Volcano plots in Figs 2B are very difficult to read. Consider changing color scheme. Are the differences noted in complex I subunits similar regardless of readout (proteomics vs immunoblot)?

Referee #2:

This manuscript by Olahova and Guerra et al. describes how the loss of function of a mitochondrial disease-related protein, RTN4IP1, produces a mitochondrial complex I (CI) assembly defect and, possibly, a deficiency in coenzyme Q (CoQ) biosynthesis.

In the title, the abstract and throughout the manuscript, the Authors state that the protein is somehow involved both in CI biogenesis and CoQ biosynthesis. On the one hand, the exact nature of the CI assembly defect produced by the absence of RTN4IP1 has been thoroughly characterized in two different human cell models, allowing the Authors to exactly pinpoint the stage in the CI assembly pathway at which the protein exerts its role. On the other hand, the direct involvement of this same protein in the CoQ synthesis pathway has not been convincingly demonstrated.

Major points:

- The Authors state that RTN4IP1 plays an "essential role" in the last stages of CI assembly. However, this does not seem to be the case in the patient fibroblasts (completely lacking the protein), where the CI structural and functional defect is much milder than in the RTN4IP1KO U2OS osteosarcoma cells, with barely any reduction in the oxygen consumption rates in the fibroblasts. Two main questions arise: why would there be such a difference in the impact of RTN4IP1 loss in the two cell lines? And, why the choice of the U2OS cell type for this study? The Authors do not discuss or justify any of these two points.
- It is true that in both defective cell lines, there is clearly a CoQ defect (decreased amounts of CoQ10 and accumulation of PPHB10), which is again more pronounced in the RTN4IP1KO U2OS cells than in the patient fibroblasts. However, the possibility that this observation could be due to a stalling of CoQ synthesis due to a secondary effect provoked by the CI defect, has not been considered. One could argue that if CoQ10 remains mostly oxidized due to a lack of reduction by CI (the main electron source), this could result in negative feedback to some of the elements involved in CoQ synthesis. This possibility is unknown and remains to be explored.
- The possibility of a secondary effect is plausible because, in the proteomics analyses, whereas the levels of CI structural subunits are clearly reduced in the KO cells, there is no mention of any of the COQ proteins being lower than in the control cells. I assume this is because none of them were found reduced in these analyses.
- This contrasts with the situation found in cells defective for PYURF (Rensvold et al, 2022) in which the protein levels of COQ5, COQ7 and COQ3 were reduced to a similar extent as several CI structural subunits and assembly factors. The Authors state that RTN4IP1 would be the second protein with a dual role in CI assembly and in CoQ synthesis after PYURF. In the case of PYURF, the justification of this dual role is convincing, as it is binding to and stabilizing several proteins with common features (S-adenosylmethionine-dependent chaperones), including COQ5, COQ3 and NDUFAF5. The last protein, even though it belongs to the family of seven- β -strand S-adenosylmethionine-dependent methyltransferases, functions to hydroxylate NDUFS7 to permit CI assembly (Rhein et al, 2016, PMID: 27226634)
- In this manuscript, what function of RTN4IP1 would allow it to perform a role both in CI assembly and in CoQ synthesis, is not clarified. In the paper by Park et al, 2023, there are experiments showing that RTN4IP1 functions as an NADH- and NADPH-dependent oxidoreductase somehow assisting COQ3 for the O-methylation of DMeQ2. In this manuscript, there is not discussion as to whether this function could be compatible with their own observations. In fact, the connection the Authors have observed is the possible interaction of COQ9 with RTN4IP1 found in (Floyd et al, 2016). In the supplemental data of the latter report, COQ9 is found to interact with more than 100 different mitochondrial proteins. Being an important point of the manuscript, this functional connection needs to be clarified experimentally or at least clearly and thoroughly discussed in the manuscript.
- As stated before, the demonstration of a function of RTN4IP1 in CI assembly is very convincingly demonstrated, and it is stated by the Authors in a much more authoritative manner than the possible additional role in CoQ synthesis. This is supported by the fact that in (Park et al, 2023), the respiratory defect caused by lack of RTN4IP1 was not rescued by CoQ2 supplementation. This, as also stated by the Authors, indicates a secondary role in CoQ biosynthesis with respect to the role in respiratory chain biogenesis, specifically in CI assembly.
- Moreover, in (Park et al, 2023) the CoQ deficiency caused by lack of RTN4IP1 was not completely restored by expression of the WT protein in a KO background. This is exactly the same finding as in the present manuscript (Figure EV1B). The authors interpret it as the demonstration that RTN4IP1 possesses a physiological role "in supporting efficient CoQ10 biosynthesis". However, the reasons why the rescue is only partial are not addressed, and it is not clear whether there was a complete rescue of CI assembly and function in the KO cells stably expressing RTN4IP1.

- The results and also the Authors' interpretation when reading the manuscript, support the observation that TMEM126A is an assembly factor of the ND4-module of CI, as published by Formosa et al, 2021 (cited in this manuscript) and by D'Angelo et al, 2021 (PMID: 33882309; not cited in this manuscript). This contrasts with the proposal in a more recent paper (Poerschke et al, 2024; cited in this manuscript), where they assign a role for TMEM126A in the insertion of mitochondrial translation products into the inner membrane, but no specific role for the assembly of CI. This point would deserve a discussion in this manuscript.
- The fact that it is still not clear whether TMEM70 is a CI assembly factor or a protein needed for CV biogenesis, should also be discussed in view of the results shown in this manuscript.

Minor points:

- Page 4: It is now known that CI is made of 44 different subunits and not 45, although it is 45 proteins in total because there are two copies of NDUFB1 (Vinothkumar et al, 2014 PMID: 25209663; Balsa et al, 2012 PMID: 22902835).
- Page 4: When talking about MITRAC15, which is also called COA1, two other references describing its role in CI biogenesis instead of in CIV assembly, as originally proposed, should be cited together with Wang et al, 2020. These are: Formosa et al, 2020 (PMID: 32320651) and Guerrero-Castillo et al, 2017.
- Page 5: "Fully assembled CI can form respiratory supercomplexes with complexes III₂ and IV, ...". Also, as mentioned later in the manuscript, non-fully assembled CI can interact with complexes III₂ and IV (Fang et al, 2021 PMID: 33852835; Moreno-Lastres et al, 2012 PMID: 22342700; Calvaruso et al, 2012 PMID: 21965299; Reviewed in Fernandez-Vizarra & Ugalde, 2022).
- Page 5: Talking about CI deficiencies, a paper from 2012 and another one from 2016 are cited. It would be convenient to cite more recent literature.
- Page 6, second paragraph, fourth line: there is a "Figu" that must not have been erased when rephrasing that sentence.

Referee #3:

Olahova and colleagues present a detailed analysis of RTN4IP1, a gene identified in patients with isolated Complex I deficiency, as a new Complex I assembly factor. This is an important and significant study given recent conflicting and high profile reports on the function of this protein. It is nice to see the results pointing toward it having a dual role in CI assembly and CoQ biosynthesis. The study is well controlled and executed, is very well written and easy to follow. For the CI assembly aspects, the authors utilised both fibroblasts from one of the original patients and a knockout cell line, analysing them using respirometry, BN-PAGE, and complexome analysis, all appropriate and widely used tools in the analysis of CI assembly. Other appropriate controls (eg. complementation of the KO) are included.

I only have two major comments and a small number of minor suggestions that may improve readability of the manuscript.

Major comments

-The experiments focused on CI assembly are overall very supportive and consistent with RTN4IP1 playing a genuine role in this capacity. The accumulation of PPHB10 in both patient fibroblasts and the KO is also consistent with and supports recent results of another group on the involvement of RTN4IP1 in CoQ biosynthesis. The dual function of RTN4IP1 in CoQ and CI biogenesis were nicely addressed in the discussion, which included reference to relevant literature. However, I'm wondering if there were any changes in migration of CoQ biosynthesis/complex Q subunits in the authors whole cell proteome or complexome data (especially in the KO where the phenotype is most pronounced)?

-While the data strongly supports RTN4IP1 having a role in CI assembly and the complexome analysis is consistent with the protein being important for the association of the ND5 module with the rest of the complex. Did the authors investigate interactors using the RTN4IP1-FLAG rescue cell line? Evidence points to transient association of RTN4IP1 with the ND5 module components, so I agree that IP from total cells using RTN4IP1-FLAG may not produce ND5 module interactors. However, it would be interesting to know if radiolabelled and newly translated ND5 can be precipitated using RTN4IP1-FLAG (as is the case with CI assembly factors associated with this module)?

Minor comments

-Fig. EV1: The expression of RTN4IP1 in the rescue looks very high. Is this the suspected reason why rescue was incomplete? It would also be nice to have the KO and rescue on the same gel as the WT so as to be able to assess how well rescue complements reduction in complex I subunits seen in the KO.

-Fig. 2A-B: the data and labels are very hard to see on these volcano plots, especially for yellow, orange, light green and light blue elements, which are essentially not visible on the manuscript I received. The size of the figures overall could also be increased (this will be an even bigger problem when typeset). I think these results could be more clearly conveyed using either heatmaps grouped into complexes/CI modules or by plotting the relative changes for each subunit to the control mean, grouped into complexes/CI modules.

-Fig. 2C (and other BN-PAGE westerns): it would be helpful if the antibody used is indicated on the figure, similar to how it is done for the SDS-PAGE westerns. Likewise in Figs. 2B and E it would be helpful to label the gels on the figure with something descriptive like 'Coomassie', 'CI NADH', 'CIV heme'.

-Figs. 2F, 3 and 4 (and linked EV): the fonts used to label some of the complexes on complexome data (eg. Lint) are almost illegible on my copy of the manuscript. Suggest to increase the size for readability. Some of the colors used to highlight regions

of the data are also a bit confusing (eg. the purple box looks pink while the dark blue box appears purple to me?). Not a big issue as you also have the protein names on hand, but potentially confusing. Fig. 4 also uses abbreviations for modules that are not used in text when first relevant eg. ND2-A when the MCIA complex is introduced on p10. Suggest to define these on first relevance in text.

-End of p6 about 1/3 of the way through the last paragraph there is a typo 'Figu

Response to Reviewer's and Editors comments
EMBOJ-2024-119510**Referee#1:**

Built from 45 known subunits coming from two genomes and containing 7 iron sulfur clusters, respiratory complex I assembly is a complicated modular process that is assisted by at minimum 18 known assembly components. Cast in this light, it is unsurprising that complex I deficiencies account for an estimated ~30% of cases of primary mitochondrial disease. It was previously shown that disease-associated RTN4IP1 variants result in isolated complex I deficiency. At the time of this initial finding, RTN4IP1 was classified as a mitochondrial protein of unknown function although it was identified in a CRISPR screen as important for oxidative phosphorylation. Here, Oláhová et al confirm that complex I expression and assembly is disrupted in RTN4IP1 variant patient-derived fibroblasts and newly generated RTN4IP1-knockout U2OS cells and formally establish that this decrease in expression and assembly results in a significant drop in complex I-linked respiration. Using complexome profiling, evidence is provided that RTN4IP1 is a complex I assembly factor that operates at a late stage of assembly, suggested to reflect a defect in the engagement of the ND5 and ND4 modules which results in a decreased abundance of N module components that fail to assemble in high molecular weight supercomplexes. Based on a recent paper that established that RTN4IP1 is a matrix-localized NADPH oxidoreductase that promotes the activity of COQ3, CoQ levels were shown here to be decreased in patient-derived fibroblasts and RTN4IP1-KO cells which also accumulated the CoQ precursor, PPHB10. While the results largely support the conclusions being made, they are, with the exception of the complexome profiling, confirmatory in nature and lacking in mechanistic insight. For example, while the complexome profiling does clearly indicate that complex I assembly is impaired, likely at the step of ND5 module docking onto ND4, the fact that RTN4IP1 does not co-migrate in an informative manner leaves it very much unclear as to how it contributes to this process. And while the twin roles in complex I assembly and CoQ biosynthesis are intriguing, no effort has been made to determine if these processes are linked or separate.

Major Points

1. Some effort to determine mechanistically how RTN4IP1 promotes complex I assembly is needed. The current discussion section nicely outlines potential productive avenues to pursue.

Response: We thank Reviewer 1 for taking the time to evaluate our manuscript and for their helpful comments. We have now expanded our experimental dataset to gain additional mechanistic insight into the function of RTN4IP1 in complex I assembly, performing further analyses of the proteomic and complexome profiling data to further clarify how RTN4IP1 promotes complex I assembly and to identify RTN4IP1 interacting partners.

2. In surveying the literature, I did not notice if it was known as to whether defects in complex I assembly impact CoQ biosynthesis and/or vice versa. Is CoQ biosynthesis perturbed in the absence of other complex I assembly factors? Is complex I assembly impaired when CoQ biosynthesis is disrupted in general?

Response: We recently undertook and published a large-scale multi-omics study in an attempt to assign functions to uncharacterized mitochondrial proteins, in which we performed proteomic, lipidomic and metabolomic profiling of >200 knockout cell lines covering mitochondrial genes of both known and unknown function (Rensvold *et al. Nature* 2022; PMID: 35614220; termed the "MITOMICS dataset"). This KO collection included deletions of genes encoding for CoQ biosynthetic enzymes as well as complex I structural subunits and

assembly factors, allowing us to interrogate any reciprocal relationship between CoQ and complex I deficiencies. While the COQ KO lines exhibited an expected significant loss of total CoQ₁₀ levels, no perturbation of CoQ₁₀ levels was observed in the CI subunit/assembly factor KOs (**Fig. 1A**). Additionally, analysis of the proteomics data for the COQ KO cell lines (COQ2, COQ7 and COQ8A are shown) revealed no significant global loss of CI subunits (**Fig. 1B**). Taken together, these data strongly suggest that while CoQ biosynthesis and CI assembly are intricately linked processes, disruption of one does not automatically lead to the impairment of the other. We will address this point in the manuscript Discussion.

A) MITOMICS Lipidomics Data

B) MITOMICS Proteomics Data

Figure 1. Lipidomics (A) and proteomics (B) analyses from the MITOMICS study (PMID: 35614220) investigating the reciprocal interaction between COQ and CI proteins.

3. Is RTN4IP1 activity required for complex I assembly and/or CoQ biosynthesis?

Response: We thank the reviewer for their comment and have pursued further experiments to investigate whether RTN4IP1's activity is required for CI assembly and/or CoQ biosynthesis. The nucleotide-binding motif in the *E.coli* quinone oxidoreductase consists of the AxxGxxG sequence. A Rossmann fold is present (**GxxGxx(G/A)**) in the human RTN4IP1 domain required for the NAD(P)H binding (RTN4IP1: **GASGGVG**) (**Fig. 2**). To disrupt the NAD(P)H binding site we generated two mutant versions of RTN4IP1-Flag, by introducing a **G215F** and a double **G215F/G218R** mutation into RTN4IP1. Unfortunately, RTN4IP1 expression could not be detected in these constructs, likely due to protein instability. Therefore, we generated new rescue cell lines in the U2OS RTN4IP1^{KO} background that stably expressed two of the catalytically inactive RTN4IP1 mutants originally identified in the Park *et al.* 2024 study (PMID: 37884807) (**RTN4IP1 R103H** and **RTN4IP1 G215A**). New

Figure 5 and supplementary figure (EV4A) have now been included in the manuscript. The mutant RTN4IP1 R103H and RTN4IP1 G215A were stably expressed (Fig. EV4A). Next, we performed proteomics analysis using the Thermo Orbitrap Astral on WT control, RTN4IP1^{KO} and the re-introduced RTN4IP1 WT, R103H and G215A mutant cell lines. In Figure 5A we show the changes in complex I and COQ protein abundance in each RTN4IP1^{KO}/rescue cell line compared to the WT control. The proteomic analysis confirmed our previous findings showing that RTN4IP1^{KO} cells have increased loss of complex I proteins, consistent with the complexome profiling results (Figure 5A). Furthermore, we detected most COQ biosynthetic proteins and found that their abundances were decreased in the RTN4IP1^{KO} cell line. Reintroduction of RTN4IP1 WT protein, as well as the R103H and G215A mutants, partially rescued all of these defects. We performed BN-PAGE analysis (Figure 5C) on WT control and RTN4IP1 rescue cell lines showing a similar pattern, where we detect loss of assembled complex I in the RTN4IP1^{KO}, with partial rescue with the WT and R103H and G215A mutant proteins, suggesting that RTN4IP1's enzymatic activity may not be important for 'structural' complex I assembly.

Targeted lipidomics on WT control and RTN4IP1 mutant (G215A and R103H) rescue cell lines were performed to measure oxidised/reduced CoQ₁₀ and the CoQ pathway intermediate (PPHB₁₀) to determine whether RTN4IP1 catalytic activity is required for CoQ biosynthesis. We observed a significant loss in both oxidized and reduced CoQ₁₀ in the RTN4IP1 KO cells, along with an increase in the early intermediated PPHB when compared to WT control (Figure 5D). Both the oxidised CoQ₁₀ and PPHB₁₀ phenotypes are nearly completely rescued by WT and G215A or R103H mutant RTN4IP1. The levels of reduced CoQ₁₀ were not rescued by either the reintroduction of WT or mutant RTN4IP1. Based on previous findings by Park et al. 2014 (PMID: 37884807) who measured COQ3 activity assays, the RTN4IP1 mutants (G215A and R103H), only marginally impacted the ability of RTN4IP1 to assist in the methylation activity of COQ3, suggesting that the G215A and R103H RTN4IP1 mutants may still retain some catalytic activity or support the activity of COQ3 in a manner independent of its oxidoreductase activity. Both RTN4IP1 WT and the G215A and R103H mutants are expressed at a significantly higher level than the endogenous RTN4IP1 protein (Figure EV4A), suggesting that such an increased overexpression of a hypomorphic mutant may still be sufficient to partially rescue the Complex I assembly and CoQ phenotypes, respectively.

In summary, WT and mutant RTN4IP1 only partially rescue the Complex I defect observed in RTN4IP1^{KO} cells, as seen in both total protein abundance and assembled complex I (manuscript Figure 5A and C)). This suggests that complex I is not operating at full capacity and is therefore unable to fully reduce CoQ (Figure 5D). Measuring reduced CoQ₁₀ is technically challenging, and the predominant form that was detected is the oxidized CoQ₁₀. Reduced CoQ₁₀ represents only a small fraction of the total, therefore our sensitivity for detecting it may be limited.

sp P28304 Q0R1_ECOLI	PDEQFLFHAAAGGVGLIACQWAKA-LGAKLIGTVGTAQKAQSALKAGAWQVINYREEDLV	198
sp P28625 YIM1_YEAST	SDSKVLVIGASTSVSYAFVHIAKNYFNIGTVVIGICSKNSIERNKGLGYDYLVPYDEGSIV	225
sp Q8IPZ3 RT4I1_DROME	AHKRVLVIGGSGGVGTLAIQILKS-QKVQ-VLATCSENAIEMVRNLGADLVVDYNNPQAM	275
sp Q4W4Z2 RT4I1_CAEL	KQQRVLIHGGAGGVGSMAIQLLKA-WGCEKIVATCAKGSFDIVKQLGAI-PVDYTSQDAT	231
sp Q7T3C7 RT4I1_DANRE	AKKRVLILGGSGGVGTFAIQVMKA-WGAH-VTVTCSQNAERLVRDLGADDVVDYTAGPVE	247
sp Q924D0 RT4I1_MOUSE	KGKRALILGASGGVGTFAIQVMKA-WGAH-VTAVCSKDASELVRKLGADVEIDYTLGSVE	261
sp Q8WV3 RT4I1_HUMAN	TGKRVLILGASGGVGTFAIQVMKA-WDAH-VTAVCSQDASELVRKLGADDVIDYKSGSVE	261
sp Q0VC50 RT4I1_BOVIN	TGKRVLILGASGGVGTFAIQVMKA-WDAH-VTAVCSQDASELVRKLGADDVIDYKSGNVE	261

.: *. .: .*: : * : : . * : *

Figure 2. Protein sequence alignment of RTN4IP1 (Clustal Omega) showing human (Q8WWV3), bovine (Q0VC50), mouse (Q924D0), zebrafish (Q7T3C7), fly (Q8IPZ3), worm (Q4W4Z2), yeast (P28625) and E. coli (P28304) sequences. The red box indicates the nucleotide-binding motif.

Minor Points

1. It is argued that the ND5 module was "not able to join to the ND4-module and the large Q/ND1/ND2-intermediate." However, the ND5 module does appear to co-migrate with some proportion of the ND4 module bounded by the dashed white box. While ND5 module assembly into the S and lint complexes is clearly disrupted in the absence of RTN4IP1, I am not convinced that it is totally unable to associate with a ND4 module-containing assembly intermediate.

Response: We thank the reviewer for this insightful question, which has prompted a more detailed inspection of the submitted Digitonin-BN-PAGE complexome profiling data. We also performed analysis of the DDM-solubilized samples-BN-PAGE Complexome data generated for the WT and RTN4IP1^{KO} cell lines.

Digitonin-BN-PAGE Complexome profiling: We would respectfully disagree with the Reviewer's opinion, as we believe that comigration of ND4 and ND5 intermediates is not clearly detectable in the complexome of RTN4IP1^{KO} cells when mitochondrial membranes are solubilized with digitonin (manuscript Figure 4). Additionally, we were unable to identify NDUFB4 (ND4-module) and NDUFB2 (ND5-module). (NDUFB4 seems to play a role in supercomplex formation or stability, PMID: 38211818). Interestingly, for the ND2 module, signals from NDUFA10 and NDUFA11 were also absent. The latter is an interface subunit to complex III and is likely involved in the stabilization or assembly of respiratory supercomplexes.

DDM-BN-PAGE Complexome profiling: To further support the findings from the digitonin-BN-PAGE complexome, we examined DDM-BN-PAGE complexes, which we have now included in the manuscript and supplementary materials (**Fig. 3** below and new Figure EV1 in the manuscript). The DDM-complexome reveals comigration of the ND4 and ND5 modules; however, the ND5 module is missing from the large intermediate complex Lint, which contains Q, ND1, ND2, and ND4 modules in mitochondrial membranes from RTN4IP1^{KO} cells. The reason why the ND4 and ND5 modules initially comigrate but the ND5 module fails to join the larger Lint assemblies allows us to have a closer look at the presence of individual subunits in that part.

Interestingly in the DDM-BN-PAGE-complexome, we confirmed the absence of NDUFA10 and a severe reduction of NDUFA11 in ND2 module and NDUFB4 in the ND4 module. As expected, no hint of supercomplex formation was observed, since DDM destabilizes higher-order structures. Notably, DDM appears to solubilize intermediates from the mitochondrial membrane that were less clearly visible in the digitonin-complexome. In this context, we observed the accumulation of ND4/ND5 together with the assembly factors TMEM126A and FOXRED1. TMEM126A remains associated even after ND4 module docking to Q, ND1, ND2 intermediate. This observation contrasts with intermediates in the Digitonin-complexome where TMEM70 was bound and reflects the solubilisation conditions that shows different assembly states and intermediates.

Figure 3 (EV1): DDM-BN PAGE complexome profiling from control WT and RTN4IP1^{KO} U2OS cells.

These findings raise the question: *Why, under conditions without RTN4IP1, can the ND5 module interact with the ND4 module but fail to dock to the Q, ND1, ND2, and ND4 intermediates?*

To better understand the complexity of how the preformed **ND5 module** attaches to complete the membrane arm, we examined the recently reported structure of complex I at 2.39 Å resolution (**PMID: 37531432**, **PDB: 8OM1**). The ND5 subunit contains an exceptionally long lateral helix that extends over the ND4 module and into the ND2 module, suggesting that the late-stage assembly of the membrane arm requires a degree of structural flexibility to integrate this helix into the pre-assembled Q, ND1, ND2, and ND4 intermediate.

Complexome profiling revealed a clear absence or severe reduction of the subunits **NDUFA10**, **NDUFA11**, **NDUFB4**, and **NDUFB2** in their respective intermediate modules. **NDUFB2**, a distal subunit interacting with **ND5**, likely assembles at a late stage. In contrast, **NDUFA10**, part of the **ND2 module**, lacks direct interaction with the **ND5** helix. Interestingly, **NDUFA11**, also within the **ND2 module**, makes direct contact with **ND5** (**Fig. 4** below and new Figure 6C). These interactions are critical for the assembly and stability of Complex I. Notably, **NDUFA11** serves as an interface subunit to complex III, suggesting that its absence could hinder supercomplex formation. **NDUFB4** makes extended contacts across ND5's surface with more complex topology than **NDUFA11**. The loss of **NDUFB4**, another direct interactor of **ND5**, may further impair late-stage membrane arm assembly.

In summary, RTN4IP1 appears to play a crucial role in the late-stage assembly of the membrane arm, likely facilitating the integration of the **ND5 helix** into the preformed **Q**, **ND1**, **ND2**, and **ND4** module intermediate. The additional analysis and discussion of the complexome profiles is added to the manuscript.

Figure 4 (5C). (A) Overview of mouse complex I structure (8OM1), highlighting the subunits NDUFA11 (ND2 module), NDUFB4 (ND4 module), and ND5, which spans from the ND5 module through the ND4 module into the ND2 module. (B, C) Interface views

showing the interaction between ND5 and NDUFA11 (B) or NDUFB4 (C), with the interface regions highlighted in red and light green, respectively.

2. The attention on the absence of FOXRED1 seems arbitrary given that DMAC2 assembly is also notably altered in absence of RTN4IP1.

Response: Thank you for this comment, we now mention DMAC2.

3. For patient fibroblasts (EV4), it seems as if TMEM126A is missing from lint and FOXRED1 and DMAC2 not detected. Is this difference in TMEM126A between KO and patient cells informative?

Response: The patient cells exhibit a significantly milder assembly defect compared to the *RTN4IP1* knockout cells, with the ND5 module still being assembled. In this case, the defect is primarily reflected by a decreased amount of supercomplexes rather than a complete failure of assembly. As a result, the patient fibroblasts do not provide sufficient mechanistic insight into the role of RTN4IP1 and as such, we focused our studies on the *RTN4IP1* knockout model in order to gain a deeper understanding of the role of this protein in complex I assembly.

4. In comparison to the RTN4IP1-KO cells, only half of the ND1 module co-migrates with the CI intermediate (lint) in WT extracts. Given this seemingly major difference, it is unclear why the presence of this intermediate in WT extracts is noteworthy. Any ideas as to what this different pattern may reflect?

Response: Thank you for noting the need to clarify this point. In the wild type cells, we primarily observe the standard assembly process. Due to their high rate of cell division—and consequently increased mitogenesis and *de novo* complex I assembly—the intermediate stages are detectable, as they represent a measurable fraction under these conditions.

In contrast, the *RTN4IP1* knockout cells display an accumulation of intermediates, indicating a stalled assembly process. The delay and disruption of late-stage assembly result in an increased presence of incomplete complexes, reflecting an impairment in finalising the maturation of complex I.

5. The labels in Volcano plots in Figs 2B are very difficult to read. Consider changing color scheme. Are the differences noted in complex I subunits similar regardless of readout (proteomics vs immunoblot)?

Response: We thank reviewer for highlighting this and ensured that appropriate and visible colours were used in the final figures (manuscript 5A-B and EV4C).

Referee #2:

This manuscript by Olahova and Guerra et al. describes how the loss of function of a mitochondrial disease-related protein, RTN4IP1, produces a mitochondrial complex I (CI) assembly defect and, possibly, a deficiency in coenzyme Q (CoQ) biosynthesis. In the title, the abstract and throughout the manuscript, the Authors state that the protein is

somehow involved both in CI biogenesis and CoQ biosynthesis. On the one hand, the exact nature of the CI assembly defect produced by the absence of RTN4IP1 has been thoroughly characterized in two different human cell models, allowing the Authors to exactly pinpoint the stage in the CI assembly pathway at which the protein exerts its role. On the other hand, the direct involvement of this same protein in the CoQ synthesis pathway has not been convincingly demonstrated.

Response: We thank this reviewer for their thoughtful comments and are grateful for their positive remark noting that this work allowed us “to exactly pinpoint the stage in the CI assembly pathway at which the protein exerts its role.” We also appreciate this reviewer’s recognition that the direct involvement of RTN4IP1 in the CoQ biosynthesis pathway has not been fully explored in this manuscript. This has been mainly attributed to the recently published work by Park *et al.* (PMID: 37884807) who proposed a role for RTN4IP1 in coenzyme Q (CoQ) biosynthesis by regulating the O-methylation activity of COQ3, undertaking experiments in C2C12 myoblast cell lines originating from adult mice and activity assays with recombinant COQ3 and RTN4IP1.

To address the function of human RTN4IP1 in CoQ biosynthesis pathway using our WT and RTN4IP1^{KO} cell lines, we have now generated stable cell lines expressing mutant forms of RTN4IP1 (G215A and R103H). As described in the response above (Reviewer 1), following proteomics analyses using Thermo Orbitrap Astral analyzer for increased depth of coverage of CoQ enzymes and Complex I subunit proteins and targeted lipidomics to measure CoQ and CoQ pathway intermediates, our data suggest that the G215A and R103H mutants partially rescue the decreased complex I and CoQ protein levels, the structural complex I assembly defect and the decreased levels of oxidised CoQ₁₀ and accumulation of early biosynthetic intermediate PPHB₁₀.

Major points:

- The Authors state that RTN4IP1 plays an "essential role" in the last stages of CI assembly. However, this does not seem to be the case in the patient fibroblasts (completely lacking the protein), where the CI structural and functional defect is much milder than in the RTN4IP1KO U2OS osteosarcoma cells, with barely any reduction in the oxygen consumption rates in the fibroblasts. Two main questions arise: why would there be such a difference in the impact of RTN4IP1 loss in the two cell lines? And, why the choice of the U2OS cell type for this study? The Authors do not discuss or justify any of these two points.

Response: We thank the reviewer for this comment and the opportunity to elaborate these points further. The *RTN4IP1* patient fibroblasts harbour missense c.500C>T, p.Ser167Phe and c.806+1G>A splice site variants, which results in decreased protein abundance. If the *RTN4IP1* defect was severe, it would likely be incompatible with life, and as such these would not be transmitted through the germline. We reason that the *RTN4IP1* patient cells have adapted to cope with the defect, as it is not deleterious enough to prevent survival. Nevertheless, the patient still manifests a complex I assembly defect, albeit to a much lesser extent compared to the complete protein knockout. In addition, the complex I defect may be more pronounced in different tissues, such as muscle and brain, compared to cultured skin fibroblasts.

- It is true that in both defective cell lines, there is clearly a CoQ defect (decreased amounts of CoQ₁₀ and accumulation of PPHB₁₀), which is again more pronounced in the RTN4IP1KO U2OS cells than in the patient fibroblasts. However, the possibility that this observation could be due to a stalling of CoQ synthesis due to a secondary effect provoked by the CI defect, has not been considered. One could argue that if CoQ₁₀ remains mostly oxidized due to a lack of reduction by CI (the main electron source), this could result in

negative feedback to some of the elements involved in CoQ synthesis. This possibility is unknown and remains to be explored.

Response: We thank the reviewer for this comment and have addressed this in our response to Reviewer 1. We believe, that while CoQ biosynthesis and CI assembly are intricately linked processes, disruption of one does not automatically lead to impairment of the other, on which will elaborate further in the discussion and provide additional data in the supplementary materials from our recent large-scale multi-omics experiment, in which we performed proteomic, lipidomic, and metabolomic profiling of >200 knockout cell lines (**PMID: 35614220**) including complex I and CoQ pathways knockouts. While the COQ KO lines exhibited an expected significant loss of CoQ₁₀ levels, no perturbation of CoQ₁₀ levels was observed in the CI subunit/assembly factor KOs (**Fig. 1A**). Additionally, analysis of the proteomics data for the COQ KO cell lines (COQ2, COQ7 and COQ8A shown) revealed no significant global loss in CI subunits (**Fig. 1B**).

Our original analyses included the measurement of the oxidized CoQ levels as a proxy for total CoQ levels in the cell, as is standard in the field. We now include measurements of both oxidised and reduced CoQ levels in the manuscript (manuscript Figure 5D), showing that the levels of both are decreased following loss of RTN4IP1. Re-introducing WT and the mutant (G215A and R103H) RTN4IP1 into the RTN4IP1^{KO} background resulted in an almost complete rescue of the oxidised CoQ levels, but no rescue of the reduced CoQ levels. This is in line with the reviewer's suggestion that there is still a partial complex I defect in the RTN4IP1 rescue cells (manuscript Figures 5A and 5C), suggesting that complex I is not operating at maximum capacity and is therefore unable to fully reduce CoQ (Figure 5D) - keeping CoQ mostly in the oxidised form.

- The possibility of a secondary effect is plausible because, in the proteomics analyses, whereas the levels of CI structural subunits are clearly reduced in the KO cells, there is no mention of any of the COQ proteins being lower than in the control cells. I assume this is because none of them were found reduced in these analyses.

Response: We thank the reviewer for this comment, something we should have elaborated upon in our original submission. Our current proteomic analyses were performed on whole cell samples using an Orbitrap Exploris 240. While we achieved sufficient coverage of the more highly abundant complex I subunits, we did not detect all CoQ enzymes. This is common for lowly abundant mitochondrial proteins; as complete detection in standard proteomics experiments requires either mitochondrial enrichment or high pH fractionation. As described above, the new proteomics data set on all of the rescue cell lines using an Orbitrap Astral with unparalleled sensitivity and depth of coverage also showed a marked decrease on COQ enzyme levels in the RTN4IP1^{KO} cells.

This contrasts with the situation found in cells defective for PYURF (Rensvold et al, 2022) in which the protein levels of COQ5, COQ7 and COQ3 were reduced to a similar extent as several CI structural subunits and assembly factors. The Authors state that RTN4IP1 would be the second protein with a dual role in CI assembly and in CoQ synthesis after PYURF. In the case of PYURF, the justification of this dual role is convincing, as it is binding to and stabilizing several proteins with common features (S-adenosylmethionine-dependent chaperones), including COQ5, COQ3 and NDUFAF5. The last protein, even though it belongs to the family of seven- β -strand S-adenosylmethionine-dependent methyltransferases, functions to hydroxylate NDUFS7 to permit CI assembly (Rhein et al, 2016, PMID: 27226634)

Response: We thank the reviewer for their comment and while we state that RTN4IP1 plays a similar role to PYURF in coordinated regulation of Complex I assembly and CoQ biosynthesis, we hypothesize that the mechanisms would be distinct. We will absolutely

clarify this in the revised discussion. While it is plausible that PYURF provides concerted support of both processes by stabilizing SAM-dependent enzymes in both pathways, previous studies by Park *et al.* (PMID: 37884807) suggest that RTN4IP1 supports CoQ biosynthesis by reducing biosynthetic intermediates for optimal reactivity with their enzymes, namely COQ3. However, the mechanism by which RTN4IP1 promotes CI assembly is still elusive. Reviewer 2 brings up thematically similar points to Reviewer 1, and to address these questions, we have performed proteomics on all of the rescue cell lines, showing a partial rescue of the CI protein subunits and CI assembly by BN-PAGE analysis. In addition, we performed crosslinking affinity enrichment mass spectrometry (AEMS) on isolated mitochondria from the rescue cell lines expressing either WT or mutant RTN4IP1 and determined interacting partners of RTN4IP1, which involve subunits of complex I N-module – NDUFV2 and NDUFV3 (manuscript Figure 5E). We have now updated our Results and Discussion sections to include the data from the AEMS studies.

- In this manuscript, what function of RTN4IP1 would allow it to perform a role both in CI assembly and in CoQ synthesis, is not clarified. In the paper by Park *et al.*, 2023, there are experiments showing that RTN4IP1 functions as an NADH- and NADPH-dependent oxidoreductase somehow assisting COQ3 for the O-methylation of DMeQ2. In this manuscript, there is not discussion as to whether this function could be compatible with their own observations. In fact, the connection the Authors have observed is the possible interaction of COQ9 with RTN4IP1 found in (Floyd *et al.*, 2016). In the supplemental data of the latter report, COQ9 is found to interact with more than 100 different mitochondrial proteins. Being an important point of the manuscript, this functional connection needs to be clarified experimentally or at least clearly and thoroughly discussed in the manuscript.

Response: We thank the reviewer for raising this important point. Previously, Park *et al.* have shown that RTN4IP1 assists COQ3 in the O-methylation of DMeQ₂. We discussed this in the revised manuscript. To investigate potential interactions between RTN4IP1 and CoQ biosynthetic proteins, we performed AEMS on the rescue cell lines expressing RTN4IP1 WT and mutant proteins to capture weak or transient protein interactions that are undetectable by traditional approaches. However, we could not detect any previously reported interactions between RTN4IP1 and CoQ proteins. This may be due to technical limitations of our method, where DSSO-based crosslinking requires lysine residues within a specific proximity - it is possible that the binding interfaces did not have suitable reactive/exposed lysine residues. In addition, our method includes stringent washes to remove non-specific interactions and therefore any transient interactions may have been lost. We would like to note that the COQ3-RTN4IP1 interaction the Park *et al.* 2024 paper was identified via BioID, so technically a different and even a more sensitive approach.

- As stated before, the demonstration of a function of RTN4IP1 in CI assembly is very convincingly demonstrated, and it is stated by the Authors in a much more authoritative manner than the possible additional role in CoQ synthesis. This is supported by the fact that in (Park *et al.*, 2023), the respiratory defect caused by lack of RTN4IP1 was not rescued by CoQ2 supplementation. This, as also stated by the Authors, indicates a secondary role in CoQ biosynthesis with respect to the role in respiratory chain biogenesis, specifically in CI assembly.

Response: We appreciate the reviewer's remarks and careful attention to the study and thank for their supportive comments, particularly regarding Complex I assembly.

- Moreover, in (Park *et al.*, 2023) the CoQ deficiency caused by lack of RTN4IP1 was not completely restored by expression of the WT protein in a KO background. This is exactly the same finding as in the present manuscript (Figure EV1B). The authors interpret it as the demonstration that RTN4IP1 possesses a physiological role "in supporting efficient CoQ10 biosynthesis". However, the reasons why the rescue is only partial are not addressed, and it

is not clear whether there was a complete rescue of CI assembly and function in the KO cells stably expressing RTN4IP1.

Response: Thank you for noting the need for clarification in this conclusion. We have changed the manuscript text and Figure legend to reflect that Complex I subunit levels were also only partially rescued in the RTN4IP1^{KO} cells (manuscript Figure 5A-D). The partial rescue could be due to various 'technical' challenges related to expression levels of the lentiviral expression system/tag/cell type etc. Our latest proteomics and BN-PAGE analysis in the attempt to assess the ability of both WT and mutant RTN4IP1 to rescue the Complex I assembly defect of the RTN4IP1^{KO} cell line, still support partial rescue of the Complex I defect and full rescue of the CoQ biosynthetic defect (oxidised COQ10 and PPHB10), suggesting that these two functions of RTN4IP1 are distinct.

- The results and also the Authors' interpretation when reading the manuscript, support the observation that TMEM126A is an assembly factor of the ND4-module of CI, as published by Formosa et al, 2021 (cited in this manuscript) and by D'Angelo et al, 2021 (PMID: 33882309; not cited in this manuscript). This contrasts with the proposal in a more recent paper (Poerschke et al, 2024; cited in this manuscript), where they assign a role for TMEM126A in the insertion of mitochondrial translation products into the inner membrane, but no specific role for the assembly of CI. This point would deserve a discussion in this manuscript.

Response: We thank the reviewer for this observation. TMEM126A is frequently detected as a protein associated with late-stage assembly intermediates of the membrane arm of complex I, including ND4 and ND5 intermediates. Most complexome data have been generated using BN-PAGE as the standard method. However, under these conditions, the mitochondrial ribosome is unstable and disassembles, making it impossible to study the coupled translation and assembly process in such complexomes. In contrast, high-resolution clear native gel electrophoresis (hrCNE) may offer a solution by enabling complexome profiling that captures these combined processes more effectively (PMID: 37615582). This is something that can be addressed and further optimised in future studies. As we see TMEM126A bound to intermediates, we cannot discuss anything according to its role in translation. We do now cite the D'Angelo et al, 2021 publication in our manuscript.

- The fact that it is still not clear whether TMEM70 is a CI assembly factor or a protein needed for CV biogenesis, should also be discussed in view of the results shown in this manuscript.

Response: We appreciate the reviewer's remarks on this topic. As noted, TMEM70 was initially identified as a complex V assembly factor (PMID: 18953340) and more recently shown to play a crucial role in the rotor ring assembly (PMID: 33753518). However, several studies have also detected TMEM70 bound to late-stage assembly intermediates of complex I. Although pathogenic *TMEM70* variants have not been linked to severe complex I assembly defects, they are associated with a mild accumulation of assembly intermediates (PMID: 32275929).

Since TMEM70 consistently appears in complexome data and has been reported by others (PMID: 32275929; PMID: 33753518), we continue to highlight its presence in these intermediates.

Minor points:

- Page 4: It is now known that CI is made of 44 different subunits and not 45, although it is

45 proteins in total because there are two copies of NDUFB1 (Vinothkumar et al, 2014 PMID: 25209663; Balsa et al, 2012 PMID: 22902835).

Response: We apologise for this confusion and have amended the manuscript to clarify that Complex I is made of 44 different subunits.

- Page 4: When talking about MITRAC15, which is also called COA1, two other references describing its role in CI biogenesis instead of in CIV assembly, as originally proposed, should be cited together with Wang et al, 2020. These are: Formosa et al, 2020 (PMID: 32320651) and Guerrero-Castillo et al, 2017.

Response: We thank the reviewer for pointing this out and apologise for this oversight. We have now added the additional references.

- Page 5: "Fully assembled CI can form respiratory supercomplexes with complexes III2 and IV, ...". Also, as mentioned later in the manuscript, non-fully assembled CI can interact with complexes III2 and IV (Fang et al, 2021 PMID: 33852835; Moreno-Lastres et al, 2012 PMID: 22342700; Calvaruso et al, 2012 PMID: 21965299; Reviewed in Fernandez-Vizarra & Ugalde, 2022).

Response: Thank you for your comment, we have now reflected on this in the revised manuscript.

- Page 5: Talking about CI deficiencies, a paper from 2012 and another one from 2016 are cited. It would be convenient to cite more recent literature.

Response: Apologies for this oversight. Additional references related to Complex I deficiencies have been added to the manuscript (PMID: 34158150 and PMID: 32454403).

- Page 6, second paragraph, fourth line: there is a "Figu" that must not have been erased when rephrasing that sentence.

Response: We thank the reviewer for pointing this out and apologise for this oversight, this has now been amended.

Referee#3:

Olahova and colleagues present a detailed analysis of RTN4IP1, a gene identified in patients with isolated Complex I deficiency, as a new Complex I assembly factor. This is an important and significant study given recent conflicting and high profile reports on the function of this protein. It is nice to see the results pointing toward it having a dual role in CI assembly and CoQ biosynthesis. The study is well controlled and executed, is very well written and easy to follow. For the CI assembly aspects, the authors utilised both fibroblasts from one of the original patients and a knockout cell line, analysing them using respirometry, BNPAGE, and complexome analysis, all appropriate and widely used tools in the analysis of CI assembly. Other appropriate controls (eg. complementation of the KO) are included.

Response: We appreciate the reviewer's remarks and careful attention to our study and thank them for their very supportive comments, particularly those highlighting that this is an "important and significant study given recent conflicting and high profile reports on the function of this protein." Moreover, this reviewer finds the study "well controlled and

executed". We have attempted to address the reviewers' specific comments related to our manuscript below.

I only have two major comments and a small number of minor suggestions that may improve readability of the manuscript.

Major comments

-The experiments focused on CI assembly are overall very supportive and consistent with RTN4IP1 playing a genuine role in this capacity. The accumulation of PPHB10 in both patient fibroblasts and the KO is also consistent with and supports recent results of another group on the involvement of RTN4IP1 in CoQ biosynthesis. The dual function of RTN4IP1 in CoQ and CI biogenesis were nicely addressed in the discussion, which included reference to relevant literature. However, I'm wondering if there were any changes in migration of CoQ biosynthesis/complex Q subunits in the authors whole cell proteome or complexome data (especially in the KO where the phenotype is most pronounced)?

Response: We thank the reviewer for their comment and provide additional complexome profiling data analysis to address their question. CoQ proteins are not known to form a single uniform complex that migrates as a discrete unit on BN-PAGE and they are thought to function as more dynamics or transient assemblies, which makes their detection in complexome profiling data challenging. While RTN4IP1^{KO} cells show a decrease in the abundance of CoQ biosynthesis proteins (manuscript Figure 5A), complexome profiling data from patients with *RTN4IP1* mutations fail to detect a number of CoQ proteins (**Fig. 5 and Fig. 6** below). Overall, we cannot draw firm conclusions on CoQ protein assemblies from our complexome data.

Fig. 5: Complexome profiling of control and RTN4IP1^{KO} cells. Proteins involved in CoQ biosynthesis (pink) and the oxidative phosphorylation system are highlighted. The position and abundance of proteins involved in CoQ biosynthesis appear comparable to the control condition, suggesting that loss of RTN4IP1 does not influence the protein composition of the CoQ biosynthesis pathway.

Fig. 6: Complexome profiling of control and RTN4IP1 patient fibroblasts. Proteins involved in CoQ biosynthesis (pink) and the oxidative phosphorylation system are highlighted.

-While the data strongly supports RTN4IP1 having a role in CI assembly and the complexome analysis is consistent with the protein being important for the association of the ND5 module with the rest of the complex. Did the authors investigate interactors using the RTN4IP1-FLAG rescue cell line? Evidence points to transient association of RTN4IP1 with the ND5 module components, so I agree that IP from total cells using RTN4IP1-FLAG may not produce ND5 module interactors. However, it would be interesting to know if radiolabelled and newly translated ND5 can be precipitated using RTN4IP1-FLAG (as is the case with CI assembly factors associated with this module)?

Response: Reviewer 3 raises a similar point to Reviewer 2 regarding RTN4IP1 interactions with other proteins. To address this point, performed crosslinking AEMS on isolated mitochondria from the rescue cell lines expressing either WT or mutant RTN4IP1. This allowed us to measure weaker and/or transient interacting partners and identified interactions between RTN4IP1 and the NDUFV2 and NDUFV3 proteins, respectively. We have not identified any ND5 module interacting partners. As it has been noted by the Reviewer, it is possible that any transient interactions between RTN4IP1 and with ND5 components may be too unstable to be captured by our current methods. Radiolabelling newly translated ND5 and testing whether it co-precipitates with RTN4IP1-FLAG would be a complementary approach, however, we believe this is outside of the scope of this current manuscript.

Minor comments

-Fig. EV1: The expression of RTN4IP1 in the rescue looks very high. Is this the suspected reason why rescue was incomplete? It would also be nice to have the KO and rescue on the same gel as the WT so as to be able to assess how well rescue complements reduction in complex I subunits seen in the KO.

Response: We appreciate the reviewer's comment and acknowledge that the expression of RTN4IP1 in the rescue cell lines is indeed very high and perhaps mis-localization could account for the inability to fully rescue the RTN4IP1^{KO} phenotypes. However, it does not seem to be a case of a dominant negative effect, as overexpression of RTN4IP1 in the WT background does not have any deleterious effects on total CoQ or PPHB levels (**Fig. 7A** below) or complex I protein abundance (**Fig. 7B** below). Western blot analyses of all samples run on the same gel showing endogenous and overexpressed RTN4IP1 levels have been included in the new supplementary figure EV4A.

Fig. 7. A) Targeted lipidomic analysis on U2OS WT cells expressing RTN4IP1^{WT} or GFP control. **B)** Volcano plot of protein abundance in U2OS WT cells expressing RTN4IP1^{WT} relative to U2OS WT(+GFP) control. Complex I subunits are highlighted in blue.

-Fig. 2A-B: the data and labels are very hard to see on these volcano plots, especially for yellow, orange, light green and light blue elements, which are essentially not visible on the manuscript I received. The size of the figures overall could also be increased (this will be an even bigger problem when typeset). I think these results could be more clearly conveyed using either heatmaps grouped into complexes/CI modules or by plotting the relative changes for each subunit to the control mean, grouped into complexes/CI modules.

Response: We thank the reviewer for highlighting this and ensured that appropriate colour schemes/font/size and visualisation of the results were used and implemented within the final set of revised figures.

-Fig. 2C (and other BN-PAGE westerns): it would be helpful if the antibody used is indicated on the figure, similar to how it is done for the SDS-PAGE westerns. Likewise in Figs. 2B and E it would be helpful to label the gels on the figure with something descriptive like 'Coomassie', 'CI NADH', 'CIV heme'.

Response: Thank you, we have now expanded our description of Figures 2B and 2C.

-Figs. 2F, 3 and 4 (and linked EV): the fonts used to label some of the complexes on complexome data (eg. Lint) are almost illegible on my copy of the manuscript. Suggest to increase the size for readability. Some of the colors used to highlight regions of the data are also a bit confusing (eg. the purple box looks pink while the dark blue box appears purple to me?). Not a big issue as you also have the protein names on hand, but potentially confusing. Fig. 4 also uses abbreviations for modules that are not used in text when first relevant eg. ND2-A when the MCIA complex is introduced on p10. Suggest to define these on first relevance in text.

Response: We thank the reviewer for highlighting this and have now defined the relevant abbreviations in the text. We have also relabelled Figures where the fonts are small.

-End of p6 about 1/3 of the way through the last paragraph there is a typo 'Figu

Response: We thank the reviewer for pointing this out and apologise for this oversight, this has now been amended.

Dear Rob,

We have now received re-review reports from three referees, which I have included below. As you will see, all note that the manuscript stands much more firmly on its main findings. Having said this, Referee #1 remains concerned that the RTN4IP1 rescue is not complete, suggesting that the epitope tag might be responsible. Could you clarify whether an untagged version was also transformed in parallel? I would also like you to consider the points for discussion raised by Referee #2. In addition, there are some remaining editorial points which need to be addressed. In this regard would you please:

- acknowledge the following funding in our online submission system: Deutsche Forschungsgemeinschaft (DFG), FOR5046 grant number WI 3728/1-1 (project number 426950122), WI-3728/3-1 (project number 515944830), SFB1531 (project S01#456687919) and the German Federal Ministry of Education and Research (BMBF, Bonn, Germany) grant to the German Network for Mitochondrial Disorders (mitoNET, 01GM1906D); NIH grants R01NS112381, R01NS131322; and the NIH award R35GM131795 and funds from the BJC Investigator Program; Pathology Society and Mito Foundation; UK NIHR Biomedical Research Centre in Age and Age-Related Diseases award to the Newcastle upon Tyne Hospitals NHS Foundation, the Lily Foundation, LifeArc and the UK NHS Highly Specialised Service for Rare Mitochondrial Disorders,
- ensure all figure callouts in the text are used sequentially; include a callout for Fig. 3A; remove the callout for Supplementary Table 1 as no such table has been uploaded,
- include a 'Reagents and Tools' table,
- label Source Data figure panels in the zip folders,
- remove the AC/CrediT section from the text,
- provide specific URLs for PXD055511, PXD064861, MSV000098108 datasets in the data availability statement,
- state exact p values in the legends of figures 1C, F, G, H, I; 5D; EV4 B, state the statistical test that was used for data analysis in the legend of figure EV4 B,
- rename "Summary" as "Abstract", and
- correct the section order as follows: Title page - Abstract - Keywords - Introduction - Results - Discussion - Methods - Data Availability - Acknowledgements - Disclosure and Competing Interests Statement - References - Figure Legends - Table(s) - Expanded View Figure Legends.

We include a synopsis of the paper (see <http://emboj.embopress.org/>). Please provide me with a general summary image, a two sentence statement and 3-5 bullet points that capture the key findings of the paper.

I am looking forward to receiving your revised manuscript.

EMBO Press is an editorially independent publishing platform for the development of EMBO scientific publications.

Best wishes,

Will

William Teale, PhD
Editor
The EMBO Journal
w.teale@embojournal.org

- a point-by-point response to the referees' comments, with a detailed description of the changes made (as a word file).
- a word file of the manuscript text.
- individual production quality figure files (one file per figure)
- a complete author checklist, which you can download from our author guidelines (<https://www.embopress.org/page/journal/14602075/authorguide>).

- Expanded View files (replacing Supplementary Information)

We realize that it is difficult to revise to a specific deadline. In the interest of protecting the conceptual advance provided by the work, we recommend a revision within 3 months (28th Sep 2025). Please discuss the revision progress ahead of this time with the editor if you require more time to complete the revisions. Use the link below to submit your revision:

Referee #1:

The authors have addressed most of my points through additional analyses and experiments. In the process, they have significantly improved their study. However, one sticking point remains. The conclusion that RTN4IP1 plays distinct roles in complex I maturation and CoQ biosynthesis is in part centered on the rescue experiments with FLAG tagged WT and mutant constructs (it is acknowledged that the failure of CoQ2 to rescue RTN4IP deficient phenotypes in the Park study are consistent with this possibility as stated in the Discussion). However, the ability of the reported results to support this conclusion is hampered by the inability of WT RTN4IP1 to fully rescue the tested phenotypes.

Point

The partial rescue of assembled complex I by WT and mutant RTN4IP1 alike is interesting and does suggest that RTN4IP1 activity is not needed for this function. Given that both mutants are expressed in lower amounts than WT RTN4IP1, it would seem unlikely that the partial rescue reflects overexpression of hypomorphs, since the degree of rescue for every parameter tested is similar/the same for WT and mutant RTN4IP rescue lines. Has it been tested whether the FLAG tagged RTN4IP1 variants are localized properly in cells? Has rescue with untagged constructs been tested? Given that the WT RTN4IP1-FLAG only confers a partial rescue that is quantitatively similar to the two mutants tested, it seems possible that the added FLAG tag is disrupting an important aspect of RTN4IP1 biology. The argument that the failure to rescue reduced CoQ levels may reflect a failure to restore full complex I function via a mechanism that is possibly separate from the role of RTN4IP1 in late stage complex I assembly is appealing but not consistent with fact that reduced CoQ levels remain low even in context of reintroduced WT RTN4IP1. Thus, I am concerned by the failure of WT RTN4IP1 to fully rescue the tested phenotypes which could stem from the addition of an epitope tag. This incomplete rescue compromises the ability to strongly conclude that RTN4IP1 has separate roles in complex I maturation and CoQ biosynthesis. In the absence of additional experiments, this conclusion should be further qualified. However, obtaining a complete rescue of the reported RTN4IP1-KO phenotypes (and including data showing both complex I assembly and function as done in Fig. 1G and Fig. 2A-C) is necessary to determine if the tested mutants differ in any way relative to WT RTN4IP1.

Minor points

1. Fig. 1G: I believe that it should be CII-linked and not CIII-linked in figure.

2. Page 10: "involving at least five ND4-module subunits (NDUFB5, NDUFB6, NDUFB10 and NDUFB11) (Fig 4, light blue box)..."

Only 4 ND4 module subunits are named which seems entirely consistent with heatmaps. Suggest changing "five" to "four".

3. 2nd paragraph of Discussion: "We demonstrated that, in addition to CI deficiency, the CoQ biosynthesis pathway is also impaired in the RTN4IP1 patient fibroblasts (Fig 1G)."

I believe that the callout should be for Fig. 1H.

Referee #2:

In this revised manuscript by Olahova, Guerra et al., the evidences for the direct involvement of RTN4IP1 in CoQ10 biosynthesis are much more solid than in the original submission. However, there are still some points that need to be clarified in the manuscript before publication:

- Whether CI assembly/activity and CoQ synthesis are mutually affected by defects in either pathway was an important point

raised by two of the reviewers. Thus, it would be good to mention this in the manuscript before discussing it at the end of the paper. It would be good to include the evidence that different CI defects do not affect CoQ10 levels, and that other CoQ10 synthesis defects do not affect CI levels (Figure 1 in the point-to-point response to the reviewers). This would fit either in the main Figure 1 or as a EV Figure 1, and it could be mentioned in a paragraph after the description of the results shown in Figure 1 (showing an impairment in both CI assembly and in CoQ10 synthesis) in page 8.

- Unless the two roles of RTN4IP1 can be dissected, which was not achieved by using mutated variants in the alleged catalytic domain (see point below), the statements that the role in CI assembly and in CoQ10 are distinct should be softened. It would be possible that the same activity could affect both processes, similar to what has been observed with PYURF.

- Since the R103H and G215A RTN4IP1 variants are able to partially rescue CI activity and CoQ10 levels as much as the WT RTN4IP1, these results could be eliminated from the manuscript as they do not contribute much to the final conclusions of this paper. They only contribute as a confounding factor as they seem to contradict the findings in the paper by Park et al. 2024, as the present manuscript shows that the oxidoreductase catalytic activity of RTN4IP1 is not necessary for its function either as a CI assembly factor or as a CoQ10 biosynthetic factor.

- Whether the patient fibroblasts contain some residual amount of RTN4IP1 should be clarified. Both the original report (Charif et al., 2018) and the present manuscript state that "these specific RTN4IP1 variants led to complete loss of immunodetectable RTN4IP1" (page 6). However, in the response to the reviewers, the authors state "The RTN4IP1 patient fibroblasts harbour missense c.500C>T, p.Ser167Phe and c.806+1G>A splice site variants, which results in decreased protein abundance". Again, this is an important point that needs to be clarified in the manuscript, because if the variants led to a complete loss of the protein, the word "essential" should be eliminated from the title, and substituted by "necessary" or something similar, because the complete loss of the protein still allows for a substantial CI assembly and CoQ10 synthesis in the patient-derived fibroblasts.

Referee #3:

The authors have appropriately addressed my concerns in their revised submission. With these and the changes suggested by the other reviews, I feel this is a much stronger manuscript than the one originally submitted.

**Response to Reviewer's and Editors comments
EMBOJ-2024-119510R****Referee #1:**

The authors have addressed most of my points through additional analyses and experiments. In the process, they have significantly improved their study. However, one sticking point remains. The conclusion that RTN4IP1 plays distinct roles in complex I maturation and CoQ biosynthesis is in part centered on the rescue experiments with FLAG tagged WT and mutant constructs (it is acknowledged that the failure of CoQ2 to rescue RTN4IP deficient phenotypes in the Park study are consistent with this possibility as stated in the Discussion). However, the ability of the reported results to support this conclusion is hampered by the inability of WT RTN4IP1 to fully rescue the tested phenotypes.

Response: We thank Reviewer 1 for taking the time to re-evaluate our revised manuscript and for their helpful comments; we are pleased to hear that the additional experiments have significantly improved the study.

Major point

The partial rescue of assembled complex I by WT and mutant RTN4IP1 alike is interesting and does suggest that RTN4IP1 activity is not needed for this function. Given that both mutants are expressed in lower amounts than WT RTN4IP1, it would seem unlikely that the partial rescue reflects overexpression of hypomorphs, since the degree of rescue for every parameter tested is similar/the same for WT and mutant RTN4IP rescue lines. Has it been tested whether the FLAG tagged RTN4IP1 variants are localized properly in cells? Has rescue with untagged constructs been tested? Given that the WT RTN4IP1-FLAG only confers a partial rescue that is quantitatively similar to the two mutants tested, it seems possible that the added FLAG tag is disrupting an important aspect of RTN4IP1 biology. The argument that the failure to rescue reduced CoQ levels may reflect a failure to restore full complex I function via a mechanism that is possibly separate from the role of RTN4IP1 in late stage complex I assembly is appealing but not consistent with fact that reduced CoQ levels remain low even in context of reintroduced WT RTN4IP1. Thus, I am concerned by the failure of WT RTN4IP1 to fully rescue the tested phenotypes which could stem from the addition of an epitope tag. This incomplete rescue compromises the ability to strongly conclude that RTN4IP1 has separate roles in complex I maturation and CoQ biosynthesis. In the absence of additional experiments, this conclusion should be further qualified. However, obtaining a complete rescue of the reported RTN4IP1-KO phenotypes (and including data showing both complex I assembly and function as done in Fig. 1G and Fig. 2A-C) is necessary to determine if the tested mutants differ in any way relative to WT RTN4IP1.

Response:

Thank you for these further comments, we would like to address the following points to support the conclusions from our rescue experiments:

We have shown a full rescue of the CoQ biosynthetic defect (oxidised CoQ₁₀ and PPHB levels, Fig. 5D) by both WT and mutant RTN4IP1-FLAG constructs expressed in the RTN4IP1^{KO} cells, suggesting that the expressed constructs are fully functional and likely localised to mitochondria. As oxidised CoQ₁₀ comprises the majority of the cellular CoQ pool, this demonstrates near complete rescue of total CoQ levels by both WT and mutant constructs.

We believe that the defect in reduced CoQ₁₀ (CoQH₂) is likely a secondary effect due to partial restoration of CI in RTN4IP1^{KO} cells. Since the electron flow from CI is essential for

CoQ reduction to CoQH₂, incomplete CI rescue would subsequently impact the levels of reduced CoQ₁₀, regardless of total CoQ abundance.

We acknowledge the reviewer's comment that the epitope tag could be interfering with RTN4IP1's role in CI assembly. However, data presented in the Park et al 2024 paper use untagged RTN4IP1 in the rescue studies and also found that they were not able to rescue the OxPhos deficiency measuring oxygen consumption rates of their RTN4IP1^{KO} cells expressing untagged RTN4IP1 (see Extended Data Figure 8C of Park et al., PMID: 37884807), and suggesting that RTN4IP1 may perform other roles independent of CoQ biosynthesis: "*We also examined OCR in Rtn4ip1-KO cells expressing RTN4IP1; although there was a slight increase, the OCR did not fully recover to the level of control cells (Extended Data Fig. 8c and Supplementary Fig. 9f). This result indicates that RTN4IP1-deficient mitochondria may suffer substantial damage, which cannot be easily recovered by its subsequent expression. Future investigations should consider the potential of permanent mitochondrial damage due to CoQ deficiency and the **broader mitochondrial roles of RTN4IP1, independent of CoQ synthesis.***"

Although we have not directly tested the localisation of RTN4IP1-FLAG (C-terminus) in our cells, our data support mitochondrial localisation of the RTN4IP1-FLAG construct. We observed a full rescue of the CoQ biosynthetic defect which takes place in the mitochondrial matrix and our crosslinking AEMS experiments were performed on isolated mitochondria from the rescue cell lines providing further evidence that RTN4IP1-FLAG protein is present in mitochondria. However, we cannot exclude the possibility that some RTN4IP1-FLAG is mislocalised in the cytosolic pool. We have now acknowledged these potential caveats in the revised manuscript on page 18:

"Although, the localisation of WT and mutant RTN4IP1-FLAG constructs and was not directly assessed, the full rescue of the CoQ defect and AEMS from isolated mitochondria support the correct mitochondrial targeting of the expressed constructs."

Minor points

1. Fig. 1G: I believe that it should be CII-linked and not CIII-linked in figure.

Response: Apologies for this oversight, we have amended Fig. 1G accordingly.

2. Page 10: "involving at least five ND4-module subunits (NDUFB5, NDUFB6, NDUFB10 and NDUFB11) (Fig 4, light blue box)..." Only 4 ND4 module subunits are named which seems entirely consistent with heatmaps. Suggest changing "five" to "four".

Response: We would like to thank the reviewer for pointing this out and apologise for this oversight. We have corrected the sentence to read:

"involving at least four ND4-module subunits (NDUFB5, NDUFB6, NDUFB10 and NDUFB11) (Fig. 4, light blue box)"

3. 2nd paragraph of Discussion: "We demonstrated that, in addition to CI deficiency, the CoQ biosynthesis pathway is also impaired in the RTN4IP1 patient fibroblasts (Fig 1G)." I believe that the callout should be for Fig. 1H.

Response: Thank you, we have made this amendment in the discussion.

Referee #2:

In this revised manuscript by Olahova, Guerra et al., the evidences for the direct involvement of RTN4IP1 in CoQ10 biosynthesis are much more solid than in the original submission. However, there are still some points that need to be clarified in the manuscript before publication:

Response: We thank the reviewer for their further constructive comments and provide further clarification on each point raised below.

1. Whether CI assembly/activity and CoQ synthesis are mutually affected by defects in either pathway was an important point raised by two of the reviewers. Thus, it would be good to mention this in the manuscript before discussing it at the end of the paper. It would be good to include the evidence that different CI defects do not affect CoQ10 levels, and that other CoQ10 synthesis defects do not affect CI levels (Figure 1 in the point-to-point response to the reviewers). This would fit either in the main Figure 1 or as a EV Figure 1, and it could be mentioned in a paragraph after the description of the results shown in Figure 1 (showing an impairment in both CI assembly and in CoQ10 synthesis) in page 8.

Response: We thank the reviewer for highlighting the importance of clarifying whether CI and CoQ pathways are equally dependent. In our response, we have now incorporated these data and present them in the revised manuscript. We have modified the results section on page 8, as suggested by the reviewer, to include these findings.

“Analysis of a previously published large-scale multi-omics study (Rensvold et al., 2022) that included the knockout collection of genes encoding for CoQ biosynthetic enzymes and CI structural subunits and assembly factors, allowed us to investigate any reciprocal relationship between CoQ and CI deficiencies. While the COQ^{KO} lines exhibited an expected significant loss of total CoQ10 levels, no perturbation of CoQ10 levels was observed in the CI subunit and assembly factor KOs (EV1A). Additionally, analysis of the proteomics data for the COQ^{KO} cell lines (COQ2, COQ7 and COQ8A/B) did not cause a global decrease in CI subunit abundance (EV1B). These data indicate that while CoQ biosynthesis and CI assembly are intricately linked processes, disruption of one does not automatically lead to the impairment of the other, supporting a possible dual role for RTN4IP1 in these pathways.”

2. Unless the two roles of RTN4IP1 can be dissected, which was not achieved by using mutated variants in the alleged catalytic domain (see point below), the statements that the role in CI assembly and in CoQ10 are distinct should be softened. It would be possible that the same activity could affect both processes, similar to what has been observed with PYURF.

Response: We thank for the reviewer’s thoughtful comment on the mechanistic relationship regarding the dual function for RTN4IP1. In our manuscript, we have referred to the previously published data by Park et al. 2024, where they showed in the O-methylation assay that the RTN4IP1 mutants R103H and G215A still retain partial activity (~50–60%):

“Previously reported COQ3 activity assays (Park et al., 2024) with addition of RTN4IP1^{R103H} and RTN4IP1^{G215A}, demonstrated that these mutations only marginally impacted the ability of RTN4IP1 to assist in the methylation activity of COQ3, suggesting that RTN4IP1^{R103H} and RTN4IP1^{G215A} mutants still retain some catalytic activity or support the activity of COQ3 in a manner independent of its oxidoreductase activity.”

We now also included the following statement in the discussion acknowledging the possibility of a shared, underlying catalytic activity:

“Our findings support that these phenotypes arise from distinct biochemical functions of RTN4IP1. Nonetheless, testing additional catalytic RTN4IP1 mutants in future studies could help clarify whether NAD(P)H oxidoreductase activity of RTN4IP1 plays a direct mechanistic role in either or both pathways.”

3. Since the R103H and G215A RTN4IP1 variants are able to partially rescue CI activity and CoQ₁₀ levels as much as the WT RTN4IP1, these results could be eliminated from the manuscript as they do not contribute much to the final conclusions of this paper. They only contribute as a confounding factor as they seem to contradict the findings in the paper by Park et al. 2024, as the present manuscript shows that the oxidoreductase catalytic activity of RTN4IP1 is not necessary for its function either as a CI assembly factor or as a CoQ₁₀ biosynthetic factor.

Response: We appreciate the reviewer’s comment and as noted, our data show that the oxidoreductase catalytic activity of RTN4IP1 is not necessary for its function either in CI assembly, or CoQ₁₀ production. We believe that this is an important finding given the presumed NAD(P)H oxidoreductase activity of this protein and its potential role in redox regulation of proteins. While the data presented in Park et al. 2024 suggest that catalytic inactivation of RTN4IP1 dampens CoQ₁₀ biosynthesis, it does not completely abolish it, further supporting our findings. However, as stated in our response above, we now include the possibility that additional studies may be required to further dissect the redox activity of RTN4IP1 in these processes.

4. Whether the patient fibroblasts contain some residual amount of RTN4IP1 should be clarified. Both the original report (Charif et al., 2018) and the present manuscript state that “these specific RTN4IP1 variants led to complete loss of immunodetectable RTN4IP1” (page 6). However, in the response to the reviewers, the authors state “The RTN4IP1 patient fibroblasts harbour missense c.500C>T, p.Ser167Phe and c.806+1G>A splice site variants, which results in decreased protein abundance”. Again, this is an important point that needs to be clarified in the manuscript, because if the variants led to a complete loss of the protein, the word “essential” should be eliminated from the title, and substituted by “necessary” or something similar, because the complete loss of the protein still allows for a substantial CI assembly and CoQ₁₀ synthesis in the patient-derived fibroblasts.

Response: We thank the reviewer for the comment above and apologise for the confusion regarding RTN4IP1 protein levels in patient fibroblasts. In the manuscript (page 6) we state the following in skeletal muscle:

“We showed that these specific RTN4IP1 variants led to complete loss of immunodetectable RTN4IP1 protein, resulting in a CI assembly defect in patient skeletal muscle (Family 11 in (Charif et al., 2018)).”

Our Mass Spectrometry-based complexome profiling data in RTN4IP1 patient fibroblast (Fig. EV3) also show loss of RTN4IP1 protein, further supporting that the (c.500C>T, p.Ser167Phe) and splice-site (c.806+1G>A) variants will most likely lead to loss of RTN4IP1 protein. To reaffirm these findings please see below the western blot showing RTN4IP1 protein levels in controls (C1, C2) and RTN4IP1 (P) patient fibroblasts and skeletal muscle samples indicating that no immunodetectable protein was present in the patient. We have ensured we have not used the term “*decreased protein abundance*” in the manuscript and would like to make it clear that the milder OXPHOS defect present in RTN4IP1 fibroblasts compared to the RTN4IP1^{KO} cells could be reflected by the fact that RTN4IP1 patient fibroblasts may retain trace amounts of mutant RTN4IP1 protein that could partially support mitochondrial functions, as well as the lasting nature of the RTN4IP1 patient mutations may

have led to the activation of compensatory pathways, which are absent in the genome edited U2OS RTN4IP1^{KO} model.

Regarding the use of the term "essential" we have change this in the title to "required". We maintain this wording in the context of normal CI and CoQ₁₀ biosynthesis, as RTN4IP1-null cells display significant functional defects in both pathways.

Referee #3:

The authors have appropriately addressed my concerns in their revised submission. With these and the changes suggested by the other reviews, I feel this is a much stronger manuscript than the one originally submitted.

Response: We thank Referee #3 for their positive feedback and appreciate their comments on the improvements made to our revised manuscript.

Remaining editorial points to be addressed:

- acknowledge the following funding in our online submission system: Deutsche Forschungsgemeinschaft (DFG), FOR5046 grant number WI 3728/1-1 (project number 426950122), WI-3728/3-1 (project number 515944830), SFB1531 (project S01#456687919) and the German Federal Ministry of Education and Research (BMBF, Bonn, Germany) grant to the German Network for Mitochondrial Disorders (mitoNET, 01GM1906D); NIH grants R01NS112381, R01NS131322; and the NIH award R35GM131795 and funds from the BJC Investigator Program; Pathology Society and Mito Foundation; UK NIHR Biomedical Research Centre in Age and Age-Related Diseases award to the Newcastle upon Tyne Hospitals NHS Foundation, the Lily Foundation, LifeArc and the UK NHS Highly Specialised Service for Rare Mitochondrial Disorders.

Response: We confirm that all the funders named above have been acknowledged in the online submission system

- ensure all figure callouts in the text are used sequentially; include a callout for Fig. 3A; remove the callout for Supplementary Table 1 as no such table has been uploaded.

Response: We have carefully checked the sequential use of all figure callouts in text and have uploaded Supplementary Table 1.

- include a 'Reagents and Tools' table.

Response: We are now including a 'Reagents and Tools' table in manuscript.

- label Source Data figure panels in the zip folders.

Response: We have now labelled the Source data figure panels.

- remove the AC/CrediT section from the text.

Response: The AC/CrediT section has been removed.

- provide specific URLs for PXD055511, PXD064861, MSV000098108 datasets in the data availability statement.

Response: We have now provided the specific URLs for all three data sets below. Please note that the PXD datasets are currently hidden and will only be made public once a DOI or PMID is available, and they will be accessible via the following links:

PXD064861:

<https://ebi.ac.uk/pride/archive/projects/PXD064861>

PXD055511:

<https://www.ebi.ac.uk/pride/archive/projects/PXD055511>

MSV000098108:

<https://massive.ucsd.edu/ProteoSAFe/dataset.jsp?task=6ccae538110b4dadbc37baf9c2e566a>

The reviewer can access the datasets following login details:

PXD055511, Username: reviewer_pxd055511@ebi.ac.uk; Password: **hKzrn5Lilzyh**

PXD064861, Username: reviewer_pxd064861@ebi.ac.uk; Password: **FQ7YYWPNW4DI**

<https://www.ebi.ac.uk/pride/>

- state exact p values in the legends of figures 1C, F, G, H, I; 5D; EV4 B.

Response: The exact p values have now been stated in the following figure legends: 1C, F, G, H, I; 5D; EV4 B (renamed EV5C)

- state the statistical test that was used for data analysis in the legend of figure EV4 B.

Response: Figure legend EV4B (renamed EB5C) has been amended to include the statistical test used.

- rename "Summary" as "Abstract".

Response: The "Summary" has been renamed to "Abstract".

- correct the section order as follows: Title page - Abstract - Keywords - Introduction - Results - Discussion - Methods - Data Availability - Acknowledgements - Disclosure and Competing Interests Statement - References - Figure Legends - Table(s) - Expanded View Figure Legends.

Response: We have corrected the order of each section of the manuscript as suggested above.

We include a synopsis of the paper (see <http://emboj.embopress.org/>). Please provide me with a general summary image, a two sentence statement and 3-5 bullet points that capture the key findings of the paper.

Response: We have created a summary image of our findings and provided a two-sentence statement and bullet points that capture the key findings of the paper below.

Pathogenic variants in RTN4IP1 lead to variable neurological phenotypes hallmarked by isolated mitochondrial complex I (CI) and coenzyme Q (CoQ) deficiency. RTN4IP1 represents a rare mitochondrial protein that appears to have independent biochemical functions in CI biogenesis and CoQ biosynthesis.

- Patient-derived RTN4IP1 fibroblasts and knockout cell models show a defect in both, CI maturation and coenzyme Q production
- RTN4IP1 functions as a late-stage CI assembly factor, required for the ND5-module stability and its docking to the ND4-module
- Loss of RTN4IP1 stalls the final membrane arm assembly, preventing N-module maturation and formation of a fully functional CI
- RTN4IP1 plays dual roles in CI maturation and CoQ biosynthesis

Dear Rob,

I am pleased to inform you that your manuscript has been accepted for publication in the EMBO Journal.

Congratulations to you and your team!

Best wishes,

Will

William Teale, PhD
Editor
The EMBO Journal
w.teale@embojournal.org
